

# Detecting impacts of extreme events with ecological in-situ monitoring networks

Miguel D. Mahecha[1,2,3], Fabian Gans[1], Sebastian Sippel[1,4], Jonathan F. Donges[5,6], Thomas Kaminski[7], Stefan Metzger[8,9], Mirco Migliavacca[1], Dario Papale[10,11], Anja Rammig[12], and Jakob Zscheischler[4]

[1]Max Planck Institute for Biogeochemistry, 07745 Jena, Germany
[2]German Centre for Integrative Biodiversity Research (iDiv), Deutscher Platz 5e, 04103 Leipzig, Germany
[3]Michael Stifel Center Jena for Data-Driven and Simulation Science, 07743 Jena, Germany
[4]Institute for Atmospheric and Climate Science, ETH Zürich, Switzerland
[5]Earth System Analysis, Potsdam Institute for Climate Impact Research, Telegrafphenberg A62, 14473 Potsdam, Germany
[6]Stockholm Resilience Centre, Stockholm University, Kräftriket 2B, 114 19 Stockholm, Sweden
[7]The Inversion Lab, Tewessteg 4, 20249 Hamburg, Germany
[8]National Ecological Observatory Network, Fundamental Instrument Unit, Boulder, Colorado, USA
[9]University of Colorado, Institute for Arctic and Alpine Research, Boulder, Colorado, USA
[10]Department for Innovation in Biological, Agro-Food and Forest Systems, University of Tuscia, Viterbo, Italy
[11]Euro-Mediterranean Center on Climate Change (CMCC), 01100 Viterbo, Italy
[12]Technische Universität München, Hans-Carl-von-Carlowitz-Platz 2, 85354 Freising, Germany

*Correspondence to:* M. D. Mahecha (mmahecha@bgc-jena.mpg.de)

**Abstract.** Extreme hydrometeorological conditions typically impact ecophysiological processes of terrestrial vegetation. Satellite based observations of the terrestrial biosphere provide an important reference for detecting and describing the spatiotemporal development of such events. However, in-depth investigations of ecological processes during extreme events require additional in-situ observations. The question is if the density of existing ecological in-situ networks is sufficient for analyzing
the impact of extreme events, or what are expected event detection rates of ecological in-situ networks of a given size. To assess these issues, we build a baseline of extreme reductions in the Fraction of Absorbed Photosynthetically Active Radiation (FAPAR), identified by a new event detection method tailored to identify extremes of regional relevance. We then investigate the event detection success rates of hypothetical networks of varying sizes. Our results show that large extremes can be reliably detected with relatively small network, but also reveal a linear decay of detection probabilities towards smaller extreme events
in log-log space. For instance, networks with ≈100 randomly placed sites in Europe yield a $\geq 90\%$ chance of detecting the largest 8 (typically very large) extreme events; but only a $\geq 50\%$ chance of capturing the largest 39 events. These finding are consistent with probability-theoretic considerations, but the slopes of the decay rates deviate due to temporal autocorrelation issues and the exact implementation of the extreme event detection algorithm. Using the examples of AmeriFlux and NEON, we then investigate to what degree ecological in-situ networks can capture extreme events of a given size. Consistent with our
theoretic considerations, we find that today's systematic network designs (i.e. NEON) reliably detects the largest extremes. But the extreme event detection rates are not higher than they would be achieved by randomly designed networks. Spatiotemporal expansions of ecological in-situ monitoring networks should carefully consider the size distribution characteristics of extreme events if the aim is also to monitor their impacts in the terrestrial biosphere.



## 1 Introduction

Many lines of evidence are pointing towards an intensification of certain hydrometeorological extreme events, such as hot temperature extremes or droughts in many regions of the world over the next decades (Field et al., 2012). Consequently, much research focuses on understanding how extreme hydrometeorological events affect ecosystems and their functioning (overviews
of the state of research and concepts are given e.g. in Smith, 2011; Reyer et al., 2013; Niu et al., 2014; Frank et al., 2015). For instance, ecosystem responses could be manifested in extreme anomalies of phenology (Ma et al., 2015), biogeochemical fluxes (Frank et al., 2015), or even in altered ecosystem structure due to induced mortality (Hartmann et al., 2015).

Global analyses of geographical extents and integrated anomalies of extremes in the terrestrial biosphere reveal that only very few extremes affect large areas, while most of events are only of very local relevance (Reichstein et al., 2013). Nevertheless,
the integrated effects of extreme events may add up to global relevance. For instance, Zscheischler et al. (2014a) showed that extreme anomalies in gross primary production (GPP) explain the global inter-annual variability of the global gross carbon uptake to a large extent.

Earth observations (EO), in particular Satellite remote sensing data, encode much of the relevant information on anomalous ecosystem functioning (Pfeifer et al., 2012) relevant to ecosystem functioning (McDowell et al., 2015). Examples are observed
anomalies in e.g. soil moisture, snow cover, land surface temperature, leaf area index, or the fraction of absorbed photosynthetically active radiation (FAPAR). The consistent and contiguous spatiotemporal data coverage, and, more importantly, the fact that observations of the land surface typically integrate a plethora of processes, makes EO very attractive for detecting extremes affecting the land surface (see e.g. Sun et al., 2015).

Although EOs enable detecting extremes in the terrestrial biosphere, a deeper understanding of impacts on ecosystem func-
tioning profits from co-exploring in-situ observations (Frank et al., 2015; Babst et al., 2017). In fact, ecological in-situ networks play an increasingly important role for analyzing ecological phenomena and often provide a complementary perspective on natural phenomena to EO (Nasahara and Nagai, 2015; Papale et al., 2015; Wingate et al., 2015) and complement model analyses (Rammig et al., 2015). One prominent example is FLUXNET with its proven record of advancing our understanding of the functioning of terrestrial ecosystems (Balddocchi, 2014). FLUXNET assembles data on the turbulent land-atmosphere ex-
changes of $CO_2$, $H_2O$, and energy via the "eddy-covariance" (EC) technique (Aubinet et al., 2000, 2012) as they are collected in regional networks at country or continental scale (e.g. the pan-European Network Integrated Carbon Observation System ICOS, AmeriFlux, AsiaFlux etc.). Today many additional networks are operational or concatenating data from past campaigns. For instance, the International Soil Moisture Network (ISMN) includes a wide range of soil-moisture observations at different depth (Dorigo et al., 2011, 2013); phenological observations are collected for instance in EUROPhen (Wingate et al., 2015) or
Phenocam (Richardson et al., 2013), and one could easily extend this list.

The site distribution of the ecological in-situ monitoring networks is typically sparse in space. One obvious and common critique is that networks emerging either as voluntary associations of sites or being constructed on the basis of existing sites (naturally) cannot provide an equitable representation of the world's ecosystems (Schimel et al., 2015). And in fact geographic clustering of sites (Oliphant, 2012) as well as incoherence in the temporal continuity is problematic. But it was also shown that



the problem of network representativity for the spatiotemporal extrapolations ("upscaling" *sensu* Jung et al., 2009; Xiao et al., 2012; Tramontana et al., 2016) is relatively minor compared to the sheer size of the network (Papale et al., 2015).

The specific question we are addressing here is the following: What is the potential of ecological in-situ networks to monitor the impact of extreme events. The paper is based on three main pillars: We present an approach to detect detect extremes that are of regional relevance (introduced in section 3, after the data). This step is important, to avoid a bias toward considering extremes in high-variance regions only, and may be relevant contribution beyond our specific application. The second pillar explores if basic probability-theoretic explanations allow us to explain detection probabilities of the regional extremes. In this context we analyze a series of random networks as reference cases (all presented in the results; section 4). We then also analyze the detection probabilities in two real networks (NEON and Ameriflux) and compare these to random networks. Finally, the paper provides an outlook on how consideration of our remarks could lead to an improved quantitative network that would consider the distribution of extreme events.

## 2 Data

### 2.1 Earth observations, EO

To analyze the suitability of in-situ networks we need a catalogue of extremes as experienced by terrestrial ecosystems in the past years. To create such a catalogue of extreme impacts, we use here extreme negative anomalies of the "Fraction of Absorbed Photosynthetically Active Radiation", FAPAR. Its values are a dimensionless spatiotemporal indicator of how much solar radiation energy (in the PAR domain) is effectively absorbed by vegetation i.e. converted by photosynthesis (Pinty et al., 2009; McCallum et al., 2010).

FAPAR is considered an "Essential Climate Variable, ECV" (Global Terrestrial Observing System, 2008) as it supports a plethora of studies on the states and variability of the biosphere (*e.g.* Knorr et al., 2007; Verstraete et al., 2008) and plays an increasingly important role in the investigation of global biogeochemical cycles (in particular carbon and water fluxes). For instance, FAPAR can be conceptually related to GPP (typically estimated from EC tower measurements) by a relation of the general form $GPP = \varepsilon \times FAPAR \times PAR$, where $\varepsilon$ is some "light use efficiency", and PAR the "photosynthetically active radiation" (e.g. Monteith, 1977) and one may include other limiting factors. Consequently, FAPAR is an important basis for empirical estimates of GPP (Jung et al., 2008; Beer et al., 2010; Tramontana et al., 2016) and other relevant ecosystem-atmosphere fluxes *e.g.* evapotranspiration (ET; Jung et al., 2010) or is directly used as input to diagnostic biosphere models (Seixas et al., 2009; Carvalhais et al., 2010).

The temporal variability of FAPAR is influenced by vegetation development, but likewise encodes e.g. fire events amongst other extreme reduction of FAPAR which are assumed to have a more or less pronounced effect on GPP. Here we use FAPAR data derived by the JRC-TIP approach (TIP-FAPAR Pinty et al., 2011), available at 1 km spatial resolution. These estimates are based on the MODIS broadband visible and near-infrared surface albedo products (Schaaf et al., 2002). The sampling rate is 16 day periods and ranges from 2000 to 2014; in this study we use data covering Europe and continental US (excluding



Alaska). In the following we denote this data set as a 3D data cube

$$\mathbf{X} = \{x_{uvt} : \forall\, u \in 1,\ldots,U;\ v \in 1,\ldots,V;\ t \in 1,\ldots,T\} \tag{1}$$

where $u$ is the index of grid longitudes, $v$ the corresponding index on latitudes, and $t$ points at the different time steps. Each element $x_{uvt}$ is called a voxel and characterized by a well-defined space-time volume.

## 2.2 In-situ networks

Firstly, we create artificial random in-situ networks in order to systematically study effects of varying network sizes and as reference for the analysis of existing networks. Secondly, we analyze existing or recently established in-situ networks for their capability to detect impacts of extreme events.

As example for the latter we use networks of eddy-covariance data. Eddy covariance (Aubinet et al., 2000) is a method to continuously observe net land-atmosphere fluxes of $CO_2$ from towers sites. An eddy covariance tower can likewise observe fluxes of latent and sensible heat; latest efforts indicate that also other trace gases will be monitored systematically in the near future (Eugster and Merbold, 2015). The towers are generally equipped with a suite of micrometeorological devices and often even run additional sensors, for instance for soil moisture and soil temperature monitoring.

We considered **FLUXNET**, a global collection of eddy covariance data collected over partly up to two decades by today (www.fluxdata.org; for in-depth descriptions see Baldocchi, 2008; Balddocchi, 2014). The interesting aspect here is that FLUXNET is a bottom-up activity of regional networks, which decided to bring their data to a central repository. Hence, there is no systematic sampling design resulting in an unbalanced spatial coverage towards Central Europe and the contiguous US (Papale et al., 2015). In the US, FLUXNET is mainly composed by the regional network **Ameriflux**. In Europe, an overview of the most widely used EC can be found in the **European Fluxes data base** http://www.europe-fluxdata.eu and will partly maintained in the future by ICOS. Here, we still rely on the site distribution as it was contained in the LaThuile data set.

We also considered, The National Ecological Observatory Network, **NEON** (http://www.neoninc.org/; Keller et al., 2008b). NEON is an initiative to monitor ecosystems of the United States and was constructed using a systematic sampling design aiming at an equitable representation of the dominant ecoregions across the US. NEON has established both, permanent sites as well as sites that can be moved for dedicated studies. Comparable to Ameriflux, also NEON sites are equipped with eddy covariance towers, but also a large suite of additional instrumentation (SanClements et al., 2015), and human-based observations are recorded frequently (Kao et al., 2012).

## 3 Methods

### 3.1 Regional extreme event flagging

The question of how to define extreme events in spatiotemporal data cubes (see eq. 1) is key to the evaluation of the suitability of ecological in-situ networks. Initially, a typical approach is defining some threshold and identifying values exceeding this threshold as potential extremes ("peak over threshold"). Choosing a global threshold setting would be particularly suitable, if

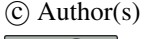



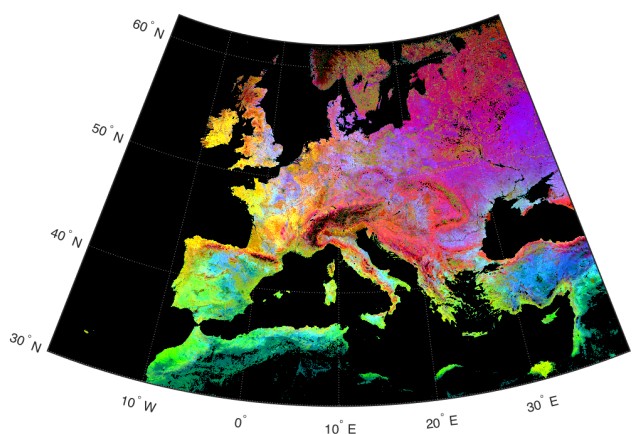

**Figure 1.** The leading three principal components of the mean seasonal cycles of FAPAR over Europe visualized as red (R), green (G), blue (B) channels. Similar RGB colours combinations indicate comparable mean phenological patterns. These similarities are exploited to define overlapping regions of comparable phenology. Within each phenological region we estimate suitable and spatially varying thresholds as reference for flagging potential extreme reductions in FAPAR.

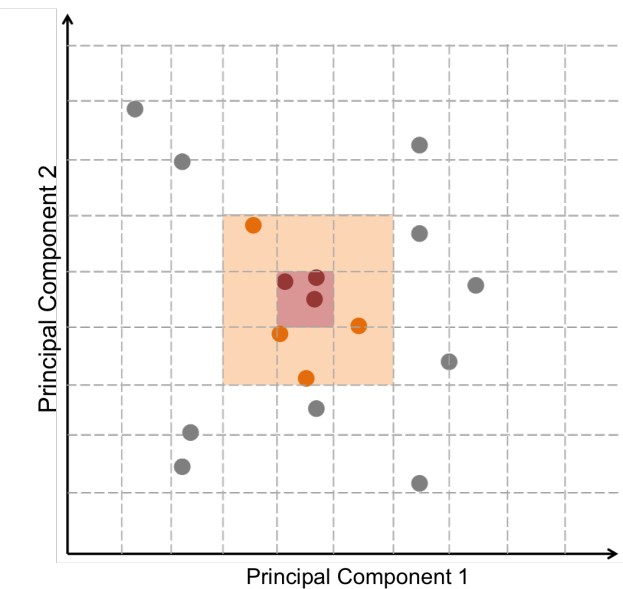

**Figure 2.** Illustration of identification of regions with similar phenology: we define a grid, where each mesh width corresponds to 4% of the total min-max range of the first EOF. Within each mesh cell, we assign percentiles as calculated over a 3×3 set of mesh elements (shown in orange) and assign these percentiles to the central dots (shown in red).

the focus is on extremes adding up to anomalies of global relevance (Zscheischler et al., 2014a), i.e. extensive data properties





where the target is the integral over space and time. However, the consequence of global thresholding is that values that are flagged as potential extremes occur exclusively in high variance regions. An alternative could be searching for local thresholds (defined over time at each $x_{uv}$). However, the latter may lead to an (implausible and undesired) equal spatial distribution of extreme event occurrences.

5    Here we develop and rely on a strategy to define thresholds of regional relevance. This is an attempt to find a compromise between fully local and global thresholding. The main steps applied for obtaining a regional threshold are the following (for a full description of the regional event detection method cf. Appendix A):

1. Estimate mean seasonal cycles of the data sets under scrutiny at each grid cell $u, v$ (centered to zero mean).

2. Compute a principal component analysis (PCA), i.e. reduce the dimensionality of the mean seasonal cycles such that each principal component (PC) represents a main gradient along the covariances of the seasonal cycles. Figure 1 shows the first three PC's as an RGB image map for Europe. We are aware that the nonlinearity of color perception by the human eye limits the quantitative informative value of the map. Still, as similar colors represent regions of similar phenological dynamics in FAPAR, one can gain an impression of environmental heterogeneity in the investigated area.

3. Identify pixels of comparable phenology by binning the three leading PC's as illustrated in Fig. 2 (note that for the sake of simplicity, Fig. 2 shows the equivalent approach when considering the first two PC's only).

4. Assign a characteristic FAPAR anomaly threshold for each time series belonging to a bin in the space of the leading PC's. Thresholds are, however, estimated from all adjacent bins in the leading PC's based on the anomalies of FAPAR (9 in case of two leading PC's, 27 in case of considering three PC's). Figure 3 illustrates the resulting regional threshold of FAPAR anomalies. In Southern European ecosystems, smaller negative anomalies of FAPAR (i.e. higher values in Fig. 3) would be used to flag values as potential extremes. The overall geographical pattern suggests that low-variance regions (i.e. arid ecosystems) require typically smaller deviations from the expected variability to be considered abnormal situations.

The rationale behind this approach is firstly, that similar mean seasonal cycles indicate which pixels form a "phenological cluster", requiring the application of similar quantiles. Secondly, the identification of these clusters based on the leading PC's avoids complications of an analogous analysis in geographical space where regions of similar phenology might be spatially separated by some barrier like a different land cover type, orography, or sea.

### 3.1.1 Contiguous spatiotemporal extremes

Based on the regional extreme threshold (Fig. 3) one may flag individual events as potential ("candidate") extremes. However, these initially flagged values may likewise reflect observational noise. Zscheischler et al. (2013) therefore proposed to consider events only as extremes if larger geographical areas are synchronously affected or if the extreme persists over some temporal instances (a very similar idea was proposed in the context of monitoring droughts by Lloyd-Hughes, 2012). This idea is realized by identifying clusters in the data cube where the spatially, or temporally voxel neighbors are likewise flagged as potential



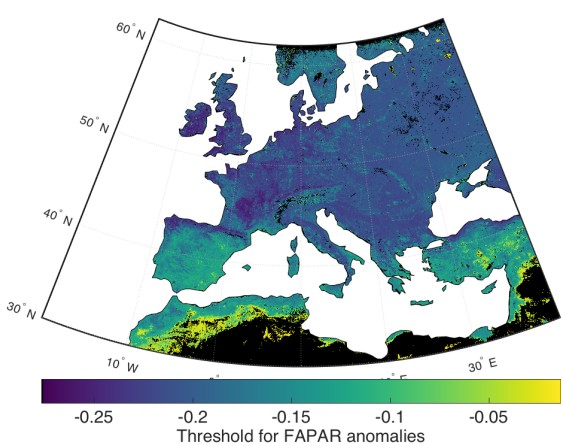

**Figure 3.** Map of the regionally varying percentiles used for detecting extreme events. The gradient between central and Southern Europe indicates that we may classify an event as extreme in one ecosystem that would be considered part of the normal variability elsewhere. Clearly, arid ecosystems have lower thresholds of extremeness in FAPAR compared to humid areas.

("candidate") extremes. Each of these clusters is subsequently considered as a singular event; for a conceptual illustration see Fig. 4.

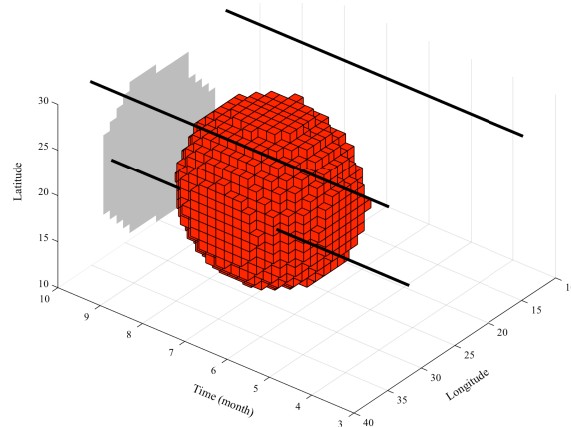

**Figure 4.** Conceptual visualization of the presented approach. An extreme occurs over a well-defined spatiotemporal domain (which could be asymmetric as shown here e.g. on the latitude/longitude projection). The rank of an extreme can be determined e.g. by the anomaly integrated by the red voxels, or the maximum spatial extent (gray area), or duration along the time axis amongst other properties (see Table 1). Black lines indicate the spatial position and active time of three in-situ measurement stations. In this example case, only one site would have coincided with the extreme and considered as potential basis for exploring the in-situ effects of the event.



**Table 1.** Properties that can be derived from the spatiotemporal extremes.

| Context | Metric | Description |
|---------|--------|-------------|
| Space | Total | Entire area effected by one extreme (not all points have to be hit instantaneously). |
| | Average | Average spatial extent. |
| | Maximum | Maximal extent affected instantaneously. |
| Time | Average | Average duration of an extreme per lat-long cell. |
| | Maximum | Maximal length of an extreme, even if the first point in time comprises different elements than the last. |
| Variable | Total | Integral of a variable over space and time. |
| | Average | Average value in the cluster of voxels considered one event. |
| | Maximum | Maximum value in the cluster of voxels considered one event. |

One critical step is defining the search space around each voxel for detecting potential neighbor extremes that should be concatenated. Throughout this paper we consider the direct neighborhood around the central voxel as follows:

– We define a spatial search space $z$. Two voxels $x_{uvt}$ and $x_{u'v't}$ ($u \neq u'$; $v \neq v'$) are connected if $|u-u'| \leq z$ and $|v-v'| \leq z$ to obtain a spatial connectivity structure for a given $t$.

5  – We also define a temporal search horizon $\tau$ from the central voxel to compare $x_{uvt}$ and $x_{uvt'}$ ($t \neq t'$) connecting them if $|t-t'| \leq \tau$.

Visually speaking, we search a square in space and a short line structure in time at the center. Note that a wide range of alternative spatiotemporal connectivity structures could be used, for instance emphasizing the temporal dimension by extending the search space along the $t$-axis. Our choice for $z = 5$ (corresponding to 25km) and $\tau = 1$ (16days) is adjusted *ad-hoc* to the

10 specific properties of the TIP-FAPAR data with its relatively high spatial resolution. By setting $z = 5$ we guarantee that e.g. similar vegetation types (from which we would assume a similar responsiveness to some extreme event) could be concatenated to one extreme, even if these vegetation types would be spatially fragmented due to a mosaic of land cover types. In time we only search starting from the central voxel, but given that we do this at each $v, u$ combinations, relatively complex spatiotemporal





structures are allowed. Each event may now consist of a set of voxels with characteristic geometric properties such as maximal duration, maximal areal extent, amongst others as outlined in Table 1. An alternative would be defining some radius in space and time around the central pixel.

### 3.1.2 Specific setting for this study

In summary, in this study we used the following settings:

- Mean seasonal cycles are computed over the time-span 2001 to 2014.

- The first three PC's are binned using a grain size of 4% of the range of the first PC.

- For each bin in the space of PC's we set a threshold that is estimated considering the anomaly values of the surrounding 26 cells.

- The search space for detecting extreme events is parameterized with $z = 5$ and $\tau = 1$ corresponding here to a search space of $\pm 5$ km and $\pm 16$ days.

## 3.2 Coinciding in-situ observations and 3D extremes

In-situ observations typically capture subgrid-level processes or footprints. For the sake of simplicity, here we assume that each point measurement is representative for one pixel $x_{uv}$ [1 km$^2$] and we intersect geographical positions $u$ and $v$ of the in-situ data with the occurrences of 3D extremes. This approach allows us answering the hypothetical question if a certain observation site would have detected an extreme in the past. An intersection considering the time domain as well, would allow us understanding if an extreme had a chance of being effectively observed. Along these lines we can also investigate if a random placement of towers would have improved/deteriorated the capacity to detect extreme events amongst other questions.

## 4 Results

### 4.1 Random networks

To better understand expectable extreme event detection rates, we initially explore random networks and their hypothetical capacity to detect extreme FAPAR reductions. We focus on Europe and vary the network sizes from $n = 5, \ldots, 10000$ sites on some logarithmic scale, asking for each size class how many of the detected extremes can be identified. More precisely, we investigate the probability that an extreme of a given size $m$ is detected by $n$ hypothetical towers $P(m, n)$. All following analyses are based on repeating the tower placement 100 times per size class. We mimic real site placement, by assuming that a tower is not mobile, i.e. it remains active at a given location over the entire period covered by the FAPAR observations. Figure 5 shows the average detection success rates for the random networks.

The ranks $r$ shown in Fig. 5a are derived here according to the spatiotemporal integrated FAPAR anomalies; the latter are on display in Fig. 5b. Across networks sizes we find that the empirical event detection probabilities increase with event impact,

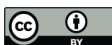


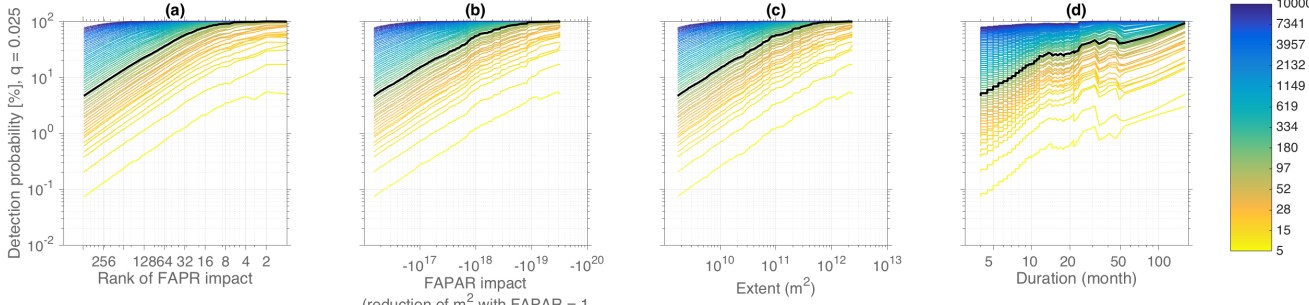

**Figure 5.** Comparison of average detection rates for randomly placed in-situ observatories of different sizes in Europe for the period 2000 to 2014. The color code shows the moderately exponentially increasing size of constantly running towers in the hypothetical random networks under consideration. Lines show the average percentage of detected events by (a) rank, (b) integrated FAPAR anomaly, (c) affected spatial area, and (d) event duration. The black line shows the case of a hypothetical network of 103 towers.

which is analogous to our findings shown before for the US in Fig. 8. These increases typically follow a straight line in the log-log plot (power-law-like behavior) for small extremes and then level off for very large event sizes. For illustrating this pattern better, we select the network of size $n = 103$ and display it as black line in Fig. 5. This specific network size detects the largest 8 extremes events with a $P \geq 90\%$. But this success rate declines rapidly for smaller events, i.e. we have only a $\geq 50\%$

chance of capturing an event of $r = 39$. While we find analogous patterns for the detection probabilities assessed in terms of spatial extents (Fig. 5c), we find no such clear pattern for the event duration. The latter is, however, explicable by the fact that we are dealing with a short spatial time series, where only few event sizes are possible.

A different view on this phenomenon is offered by Fig. A1 showing the detection likelihood for extremes of a given rank $r$ across varying networks sizes. Extremes of low rank (i.e. large in impact) need very small networks to be detected with

rates near to 100%, while high rank events (of small impact) need much larger networks to converge to similar detection rates. The detection probability scales linearly in log-log space with networks size, indicating that one would need to inflate in-situ networks by orders of magnitude to detect small scale events at comparable rates than large-scale extremes.

### 4.1.1 Statistical considerations

The results shown in Figs. 5b and 5c are an empirical approach to detecting extreme events of impact extent $m$ (measured e.g.

in terms of the number of pixels or area affected during an event) by a network of $n$ towers randomly distributed in space. These figures are reporting the probability $P(m, n)$ that *at least one* tower detects the extreme. Let us consider this in more detail for the case of "affected area" (the "key" property of extreme events in the case of FAPAR Reichstein et al., 2013). For the area-wise assessment, we begin with noting that a single extreme event of spatial extent $m$ is detected by a single randomly



placed tower with probability

$$p = \frac{m}{m_{\max}}, \tag{2}$$

where $m_{\max}$ is the maximum possible extent $m$ (in our case the maximally affected area across all time steps). However, an equivalent question is to investigate the probability that one extreme *is not detected* by any of the $n$ towers. According to the

binomial distribution, the latter probability is $(1-p)^n$, and our estimated probabilities should be described by

$$
\begin{aligned}
P(m,n) &= 1 - (1-p)^n \\
&= 1 - \left(1 - \frac{m}{m_{\max}}\right)^n.
\end{aligned} \tag{3}
$$

This formulation helps understanding the parallel decline (linear in log-log) in the detection probabilities for small extremes: We can rewrite Eq. 3 as

$$P(m,n) = 1 - \exp\left(n \ln\left(1 - \frac{m}{m_{\max}}\right)\right) \tag{4}$$

A Taylor expansion of Eq. 4 for a small number of towers $n$ and small event sizes $m/m_{\max}$ (here realized by assuming that $|\ln(1 - \frac{m}{m_{\max}})n| \ll 1$) yields

$$P(m,n) \approx -\ln\left(1 - \frac{m}{m_{\max}}\right)n. \tag{5}$$

Further approximating this formula for small extremes with $|\frac{m}{m_{\max}}| \ll 1$ then gives

$$P(m,n) \approx \frac{m}{m_{\max}}n, \tag{6}$$

which, in a logarithmic form reads as

$$\ln P(m,n) \approx \ln m + \ln n - \ln m_{max}. \tag{7}$$

We would now expect that this equation explains the empirically identified parallel lines of positive slope in Fig. 5 and compare our empirical findings to this theoretical expectation. Fig. 6 compares the expected and observed detection probabilities. Here

we note that the leveling off of event detection probabilities for large events is indeed theoretically expected, but the log-linear scaling for small events is expected to be steeper sensu Eq. 3.

In other words: the observed detection probabilities for small extremes are higher than expected, while detection probabilities of large scale extremes are lower in random networks compared to theoretical expectations. Our hypothesis is that these discrepancies are related to the spatiotemporal correlation structure of the extreme events which is not taken into account in

the above theoretical analysis.

### 4.1.2 Spatiotemporal correlations

In order to investigate the discrepancy revelad in Fig. 6 further, we perform a series of simulations using artificial data, i.e. we generated Gaussian data but introduced varying spatiotemporal correlation structures of different degrees. We followed the





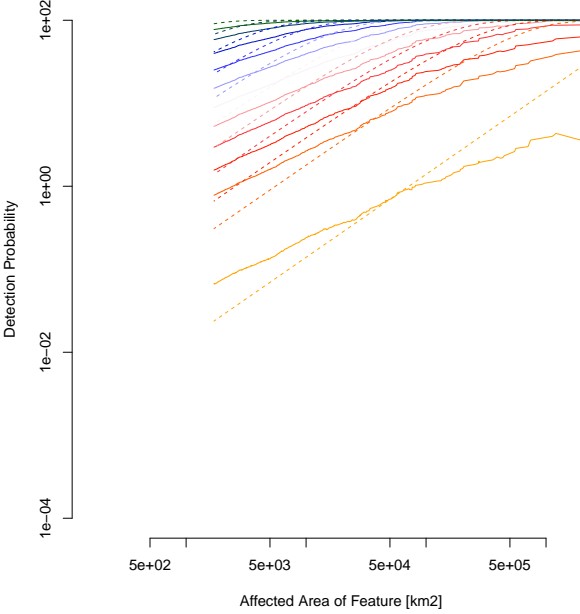

**Figure 6.** Comparison of the affected area of extremes (Fig 5c) and the theoretical expectation according to a binomial distribution and uncorrelated data. Our empirical detection probability is lower for the very large extremes, and higher for the small extremes. The deviation is more pronounced for small network sizes, while the overall structure of the expected detection probabilities is captured well by the theoretical approximation.

approach suggested by Venema et al. (2006) to simulate data with a power law power spectrum of some prescribed exponential spectral decay, i.e. $\beta = 0$ corresponds to uncorrelated, $\beta = -\frac{3}{5}$ to moderately correlated, and $\beta = -\frac{8}{5}$ to highly correlated data, as visualized in the appendix Fig. A4g-i. We used a simplified event search radius ($z = 1$, $\tau = 1$) and investigate two cases:

1. Ignoring the time domain: In this case, the empirically identified detection rates correspond exactly to the theoretical
5   detection probabilities. The findings reveal that the spatial correlation structure does not explain a deviation from the theoretically expected pattern (compare appendix Figs. A4a—c). This is explicable by the fact that, although patterns of extreme anomalies might be correlated in space, the tower placement is still random and for sufficiently sparse networks it has no effect.

2. Considering spatial and temporal correlations: In this case we find a tendency towards lower detection probabilities. This
10   effect becomes more pronounced with larger extremes and spatiotempoal autocorrelation (see appendix Fig. A4d-f). We can explain that easily, as repeated temporal observations of $m(t)$ are not independent, i.e. if an extreme is undetected at time point $t$ it likely remains undetected in the next step by the same random network. This effect must vanish for very small extremes, as the events are less correlated in time.




Yet, the approximation of the expected probabilities for the small events is still inconsistent with our empirical finding (recall Fig. 6). Hence, we repeat the artificial experiment using the exact algorithmic settings as applied to the FAPAR data: We allow for a tolerance radius ($z \gg 1$, $\tau = 1$) to identify each each extreme by a given tower. Again we distinguish the two cases:

1. Ignoring the time domain: Using a large search radius for detecting extremes (which is clearly necessary in real and e.g fragmented landscapes) leads to increasing event detection rates. This effect can lead to higher detection rates that exceed the simple statistical expectations as derived from the binomal distribution by several orders of magnitude in the case of small extremes (see appendix Figs. A5a—c).

2. Considering the full spatiotemporal case, reduces the discrepancy a bit (i.e. for large events that would be anyway detected), but still shows an overestimation (see appendix Fig. A5d-f). For the very large events, the lines may even cross in case of strongly autocorrelated data.

These numerical experiments highlight various issues that need to be considered in evaluating real networks or quantitative network-design: Firstly, the phenomena we aim to monitor are highly autocorrelated in time which leads to very few effectively independent observations. Therefore, theoretically expected detection rates estimated from the binomial distribution are overly optimistic for large events - unless the effect of autocorrelation would be analytically taken into account. Secondly, we have a severe scale issue to consider: An extreme event (e.g the 2003 heat wave) may affect large areas, but whether the local ecosystem is truly affected depends on local geographical factors.

## 4.2 Scaling issues

One doubt in applying a regional event detection approach was if key aspects of extreme event distributions would have been affected. Occurrence probabilities of extreme events in the terrestrial biosphere have been often reported to follow a power-law of the form $p(m) \propto m^{-\alpha}$ in the tails, i.e. for some values $\geq m_{\min}$ (see Reichstein et al., 2013; Zscheischler et al., 2014a, for scaling examples in FAPAR and gross primary production respectively). Using a maximum likelihood estimator as suggested by Clauset et al. (2009); Clauset and Woodbard (2013) we analyze the scaling characteristics of contiguous areas affected by extreme events. We find that the event properties follow a power law (see Fig. A3), with only marginal differences in the scaling exponents between Europe and the US: the probabilities of areas affected by extremes in Europe decline with $\alpha = 1.85 \pm 0.007$ (uncertainties given as standard errors from 1000 bootstrap samples), while in the US the scaling behaviour is characterized by a decline of $\alpha = 1.91 \pm 0.003$.

Without over-interpreting these patterns (i.e. many processes could lead to an emergence of these power-laws, some of which are discussed in Zscheischler et al., 2014b) we consider that this property could be exploited for network design issues: According to Newman (2005, and others) there are a few considerations pointing in this direction: the expectation value $E[m(r)]$ of an extreme event of rank $r$ (in this formulation, the largest event has rank 1 as in Fig. 5a) then has the form

$$E[m(r)] = cr^{-\frac{1}{\alpha-1}}. \tag{8}$$




where $\alpha$ is the scaling exponent, and $c$ some normalization constant - both can may be obtained from a fit to the empirically obtained rank function $m(r)$. Applying Eq. 8 would allow us to study the network detection probability as a function of rank (see Figs. 5a and A1) and we can insert the expressions into Eq. 3:

$$P(m,n) = 1 - \left(1 - \frac{m(r)}{m_{max}}\right)^n \tag{9}$$

$$= 1 - \left(1 - \frac{cr^{-\frac{1}{\alpha-1}}}{m_{max}}\right)^n \tag{10}$$

Further, using the approximated log-log form of the network detection probability (Eq. 8) yields

$$\ln P(m,n) \approx -\frac{1}{\alpha-1}\ln r + 1\ln n + \ln c - \ln m_{max}. \tag{11}$$

This equation may explain the numerically obtained parallel lines for ranks $r$ corresponding to small extreme event extents $m(r)$ (see e.g. Fig.A1). More importantly, it relates the scaling exponent to the expected detection probabilities.

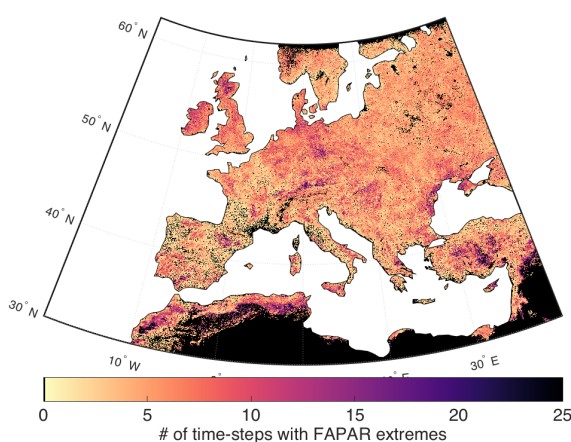

**Figure 7.** Number of FAPAR extreme events detected in Europe from 2000 to 2014 using a regional varying threshold and using a spatiotempoal detection algorithm. Moderately higher event occurrence counts are found primarily in the Maghreb, central Turkey and central Europe. However, this count is independent of the intensity and other properties of the event (as listed in Table 1), calling for a cautionary interpretation.

## 4.3 Comparing AmeriFlux and NEON

The results so far show that random networks may partly differ from our expected detection rates which has various causes. But the overarching hypothesis is that even relatively small networks may have a good chance to detect large scale extreme events. We therefore consider the configuration of the real eddy covariance networks. We now focus on the US (continental





areas only) instead of Europe as we have two networks with a quite different history, and therefore configuration: Ameriflux, NEON, and we also consider them both together. Again, we compare our results to random networks of equal size.

The starting point for our considerations was the question if ecological in-situ networks have been effectively able to detect the most relevant extreme events experienced by land ecosystems due to their network construction or if these were lucky
circumstances. We therefore ranked the largest 100 events detectable in continental US by their integrated FAPAR anomalies. We then counted the number of events that could have been detected by at least one of the Ameriflux or NEON towers, or, by taking both together (if all towers would have been active over the entire monitoring period). Fig. 8 visualizes the number of detected events for these three network configurations of NEON, AmeriFlux, as a function of their rank.

Due to its large network size, AmeriFlux detects many more extremes compared to NEON (128 vs. 39 sites in the contiguous
US, excluding Alaska and islands). Concatenating both networks helps increasing the detection rates for small events. Our next question is if this detection rates are comparable to random networks of the same size. For the case of NEON we find that the median detection rate of randomly designed networks is slightly higher compared to the real network - which still remains above the 2.5%ile. At first glance this is an unexpected finding: we would expect that undesired vicinity may occur by chance in a random network increasing redundancy amongst towers in space compared to the very systematic sampling design of NEON
(Keller et al., 2008a). We conclude here that while the design efforts to establishing NEON may pay off for certain studies, they are no effective means to maximize the detection of extremes. This observation reflects again the lack of spatial regularity in the occurrence of extremes.

The equivalent experiment realized for AmeriFlux yields much higher detection rates for the random networks compared to the established network (Fig. 8). We attribute this difference to one particular characteristic of AmeriFlux: many of the sites in
this network are co-located on purpose (e.g. to explore spatial heterogeneity or to monitor different disturbance regimes in adjacent and hence climatologically similar ecosystems). Fig. 8 basically reflects that AmeriFlux sites must have a relatively high degree of spatial clustering. If the target would be to analyze continental extreme events and guarantee monitoring the largest events, the AmeriFlux configuration could be rapidly suboptimal. In other words: the spatial autocorrelation in an ecological in-situ network that was not systematically designed can be outperformed by a random (and hence spatially independent)
network.

Another aspect to investigate in this context is concatenating NEON and AmeriFlux (both data sets are intended to be freely available to the research community, Fig. 8 dashed line). Our results show that this approach would marginally increase the detection capacity. One reason for this marginal improvement is again that AmeriFlux and NEON sites are partly geographically collocated and that AmeriFlux—despite of being a bottom-up activity—already has a significant spread across the country that
is competitive with a novel designed network for the purpose of capturing large scale extremes.





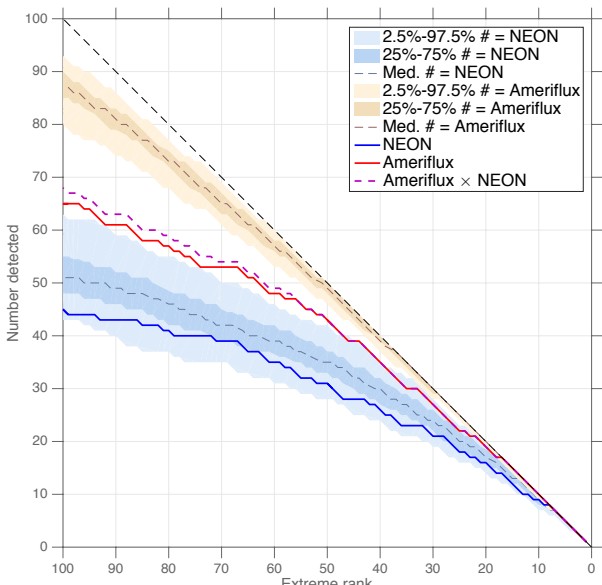

**Figure 8.** Comparison of NEON (39 terrestrial sites) and AmeriFlux (128 sites) for their potential of detecting extremes defined by varying thresholds in the continuous continental US (excluding Alaska and islands). The purple dashed line shows a merged AmeriFlux–NEON network. Remaining colored dashed lines are random networks of sizes corresponding to NEON (blue dashed, in the envelope of a 95 percentile range) and AmeriFlux (brown dashed, in the envelope of a 95 percentile range). We also show the 1:1 line, which would correspond to a perfect detection performance and is the theoretical limit.

## 5 Discussion

### 5.1 Methodological remarks

To the best of our knowledge, there are only very few realized examples of systematically designed ecological in-situ networks. One example is NEON and therefore interesting in the context of this study. The underlying design principle is to cluster environmental conditions and states, including e.g. precipitation, radiation, topography, and water table depth, amongst others (Hargrove and Hoffman, 2004). These delineated ecoregions are then representing approximately homogenous areas in the mean land-climate system state, and yields an equitable representation of land surface processes in upscaling activities (e.g. the spatiotemporal inter- and extrapolation of land-atmosphere fluxes of $CO_2$, $H_2O$, and others Jung et al., 2011; Xiao et al., 2012; Papale et al., 2015) or model-data integration studies (sensu Williams et al., 2009).

The equitable representation of ecoregions in NEON was one of the motivations for us to likewise propose an algorithm capable of detecting extremes of regional relevance. Our aim here was us identifying extremes relative to some regional reference variability. Other detection methods are relying on global thresholds. Using such a method would fundamentally change the obtained picture as it typically leads to a few hotspots of extremes in high-variance regions (for the case of GPP see Zscheischler et al., 2014b). The effect of building on a regional threshold to delineate which anomalies should be considered

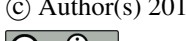



"extreme" (recall Fig. 3) is that we find only a very moderate geographical clustering of event occurrences (Fig. 7). This effect implies that we cannot really use the detected spatial patterns as priors to e.g. propose a specific sampling strategy that would aim for a more dense tower placement in certain regions.

The other aspect to discuss regarding a regionalization is that we rely on a very limited number of events detected in a finite time horizon of available satellite data. Monitoring 15 years of extreme events probably does not allow us to conclude anything about the future occurrences of extreme events. In this sense, this study can only be read as a call for (re)considering the density of a ecological networks in network design studies. An alternative would be to consider also climate projections and putting an emphasis on more "vulnerable" ecoregions. Besides of non-stationary climate and environmental conditions, we have to acknowledge that extremes are too rare to derive a spatial occurrence probability using data from the satellite era only.

Our finding that concatenating NEON and AmeriFlux would have yielded only minimal additional detection capacities for extreme events can be understood as a call to avoid co-locating towers in relative vicinities - at least if the objective of detecting extreme events would be considered highly relevant.

Prototyping an analogous study for the ISMN using a drought indicator (see Appendix B), stresses the importance of keeping network measurements alive over time. In this particular case we find that in particular small extremes (i.e. those which are typically of small spatial extent and duration) have a slightly higher detection probability compared to random networks. But the key issue is that many of the sites have been only active for short monitoring periods leading to substantial loss in event detection rates. Only continuous measurement campaigns will substantially improve event detection rates.

## 5.2 Relevance for network design

In this paper, we offered a few remarks on the network design - if the objective was only to monitor and understand the impacts of climate extremes on ecosystems. In the overall view, we showed that basic probability theoretic expectations should be be taken into account, but would need to be extended to consider temporal autocorrelation as well as event detection approaches chosen. In our case, the latter had a relatively large footprint ($z = 5$) in order to not miss events that may appear fragmented due to e.g. heterogeneous landscape characteristics. Clearly, one would need to determine such parametric choices depending on the type of extreme events and underlying question.

Nevertheless, we think that the remarks presented here could become useful elements to quantitative network design studies. In our area, earlier considerations in this direction have put their emphasis on reducing the uncertainties for upscaling fluxes from the site level to continental or global flux fields (Papale et al., 2015). Focussing on these first-order question is of course essential, before focussing on detecting rare anomalies. This is also reflected in the alternative methodological avenues that were used for addressing the network design problem. For instance, carbon cycle data assimilation systems (CCDAS; Rayner et al., 2005) were very useful for quantitative network design (QND; see, e.g. Kaminski et al., 2010) i.e. to evaluate real or hypothetical candidate networks in terms of their constraint on target quantities of interest. The QND approach within a CCDAS allows to combine terrestrial, atmospheric and ultimately also oceanic data streams. A key finding so far was that eddy covariance networks with one site per ecosystem type achieve an excellent performance. QND studies have also been performed for EO data streams such as column integrated atmospheric $CO_2$ Kaminski et al. (2010). But again, none of these





studies so far considered to unravel the impacts of extreme events on the terrestrial biosphere which might be a relevant aspect to be pursuit by subseuquent studies.

## 6   Conclusions

This study tries to understand to what degree ecological in-situ networks such as AmeriFlux or NEON can capture extreme
events of a given size that hit land ecosystems. We find, for instance, that the largest 10 extremes having occurred in the US between 2000 and 2014 would have all been identified with the current networks, offering a good perspective for in-depth site level analyses of these phenomena. Concretely, this finding means that there is a high chance of capturing the major extreme events – beyond the very few prominent 2-3 events that may receive major media coverage such as the 2003 heatwave in Europe or the 2012 US drought. In general, we find that "large" extreme events could have been detected in a very reliable
way, while there was a linear decay of detection probabilities for smaller extreme events in log-log space. We can explain this general behavior with straighforward considerations in probability theory, but the slopes of the decay rates deviate: While we find lower detection rates for the very large extremes, the opposite is the case for very small extremes. Experiments with artificial networks reveal that these deviations stem both from the temporal autocorrelation and the exact implementation of the detection algorithm.

Our original motivation for pursuing this study is the question if one could optimize the design of ecological in-situ networks for maximizing the detection rates of extreme events. And indeed, we find some general rules, i.e. when the goal is detecting very large events (i.e. low rank events), network sizes can differ by up to two orders of magnitude but still yield nearly comparable detection rates. Only if the goal was to reliably enhance the detection probabilities of small-scale events, a disproportionate "investment" in large networks would be required, but then also become orders of magnitude more efficient
compared to the small networks.

However, any inference on the future spatial occurrence probability of extremes is not tenable based on data from a decade of observation. But it is not only data paucity that limits our insights here: quantitative network design is per se non-trivial in a changing world. We find, however, that certain general patterns could be taken into consideration for instance the fact that event occurrence probabilities are clearly inversely related to detection probabilities on a very well defined and a robust scale.
Also the power-law distribution of extreme event size seems to have practical relevance for network design purposes.

*Author contributions.* The first three authors equally contributed to analyses presented in this study, J.F.D. helped in deriving the probability-theoretic explanations for the identified patterns, all authors provided substantial input to the design of the study and discussion of the results.

*Acknowledgements.* This study was supported by the European Space Agency with the Support to Science Element STSE "Coupled Atmosphere Biosphere virtual LABoratory project CAB-LAB" see http://earthsystemdatacube.org/ and we thank the EU for supporting the BACI
project funded by the Horizon 2020 Research and Innovation Programme under grant agreement 640176; D.P. further thanks the EU for



the ENVRIplus project funded in the same programme under grant agreement 654182. The National Ecological Observatory Network is a project sponsored by the National Science Foundation and managed under cooperative agreement by Battelle Ecology, Inc. This material is based upon work supported by the National Science Foundation under the grant DBI-0752017. Any opinions, findings, and conclusions or recommendations expressed in this material are those of the author(s) and do not necessarily reflect the views of the National Science

5   Foundation. J.F.D. thanks the Stordalen Foundation (via the Planetary Boundary Research Network PB.net) and the Earth League's EarthDoc program for financial support.





## Appendix A:  Supplementary figures

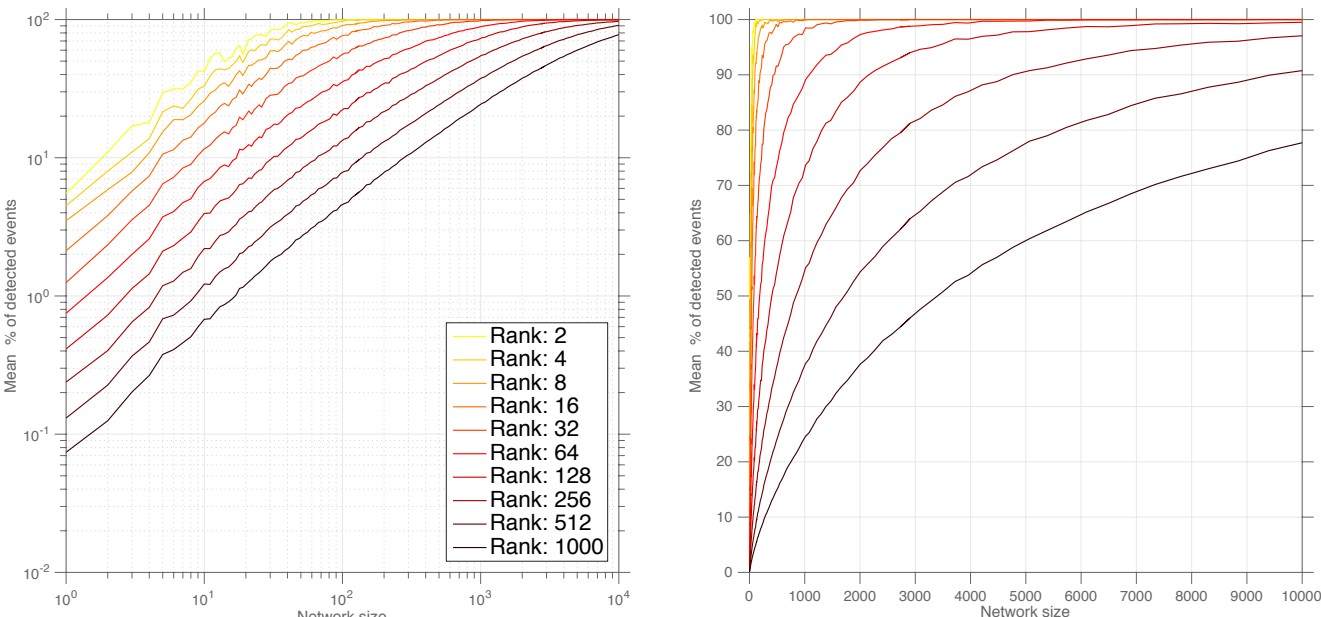

**Figure A1.** Average detection rates of extremes of given ranks (each line represents the rank of an extreme event) across varying network sizes in logarithmic representation (left panel) and linear representation (right panel). Small ranks indicate large impact extremes that typically also affect large areas (see Fig. 5). The figure shows that detection rates scale with smaller network sizes and then tend to saturate i.e. we find a converge towards full detection rates.





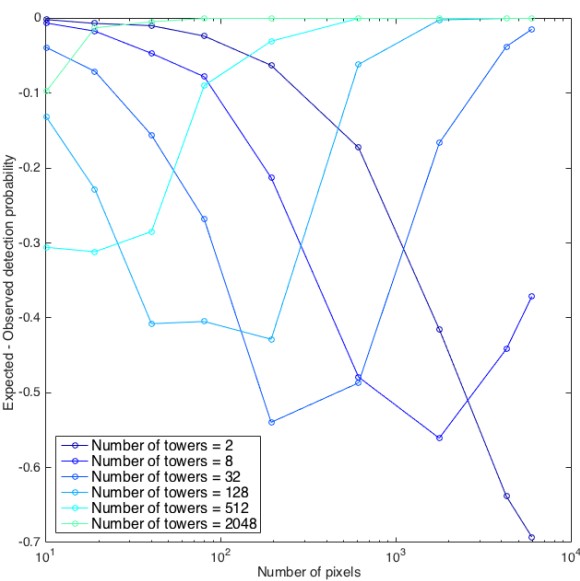

**Figure A2.** Comparison of the affected area of extremes (Fig 5c) and the theoretical expectation according to a binomial distribution. Our empirical detection probability is lower fore the very large extremes, and higher for the small extremes. The problem is more pronounced for small network sizes.





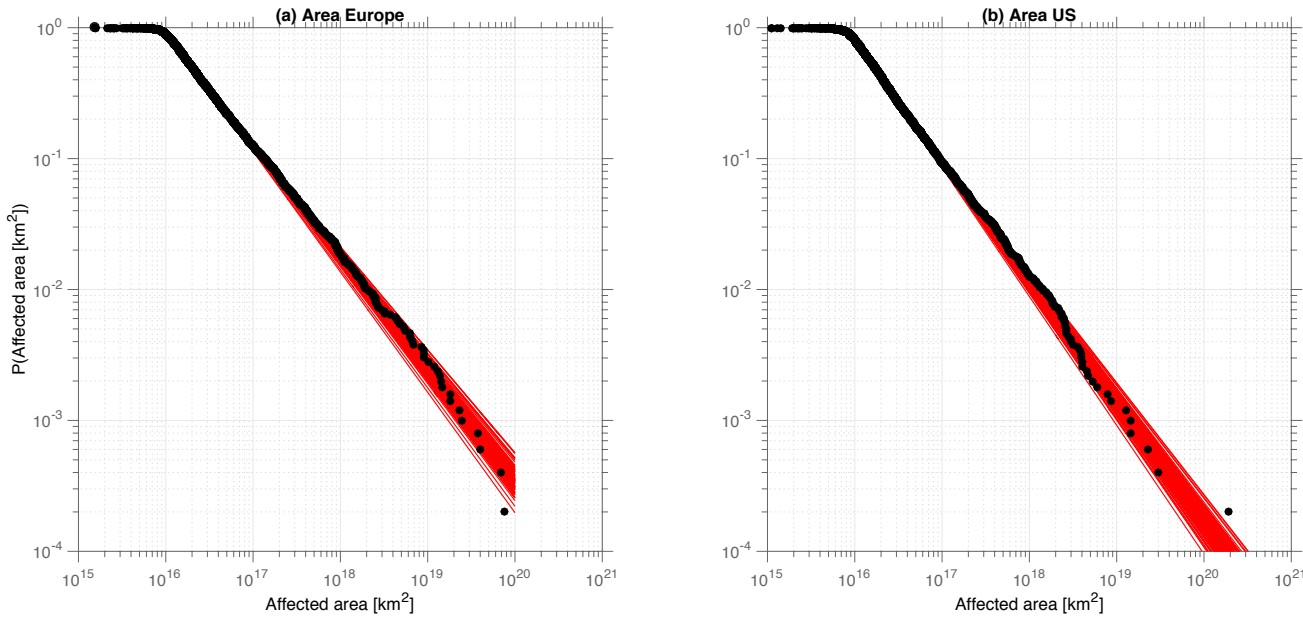

**Figure A3.** The probability distribution of areas affected by extremes in (a) Europe and (b) the US. The tails of the distributions can be described by power laws. The average scaling exponent for the tails are 1.85 for Europe and 1.91 for the US.

.





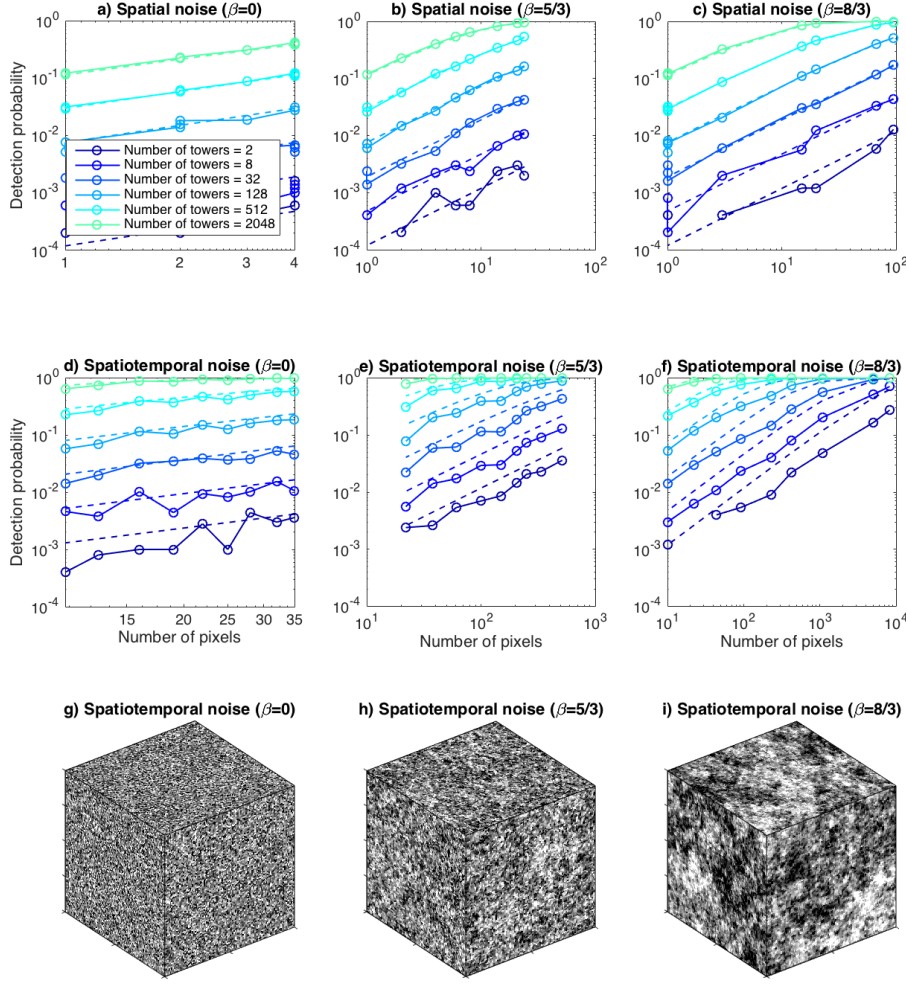

**Figure A4.** Artificial data example. a) Detection probabilities when ignoring the time domain for varying network sizes. In this case, the empirically identified detection rates correspond exactly to the theoretical detection probabilities. If we induce spatiotemporal correlations in b), and stronger in c) we still find a excellent it to the theoretical expectation as we still have relatively sparse networks and the towers are independent samples of the underlying distribution. If the detection rates over space and time are considered, however, the events are not anymore independent due to their temporal autocorrelations and e) and f) reveal lower detection rates. g), f), i) are the data corresponding to results in the columns.




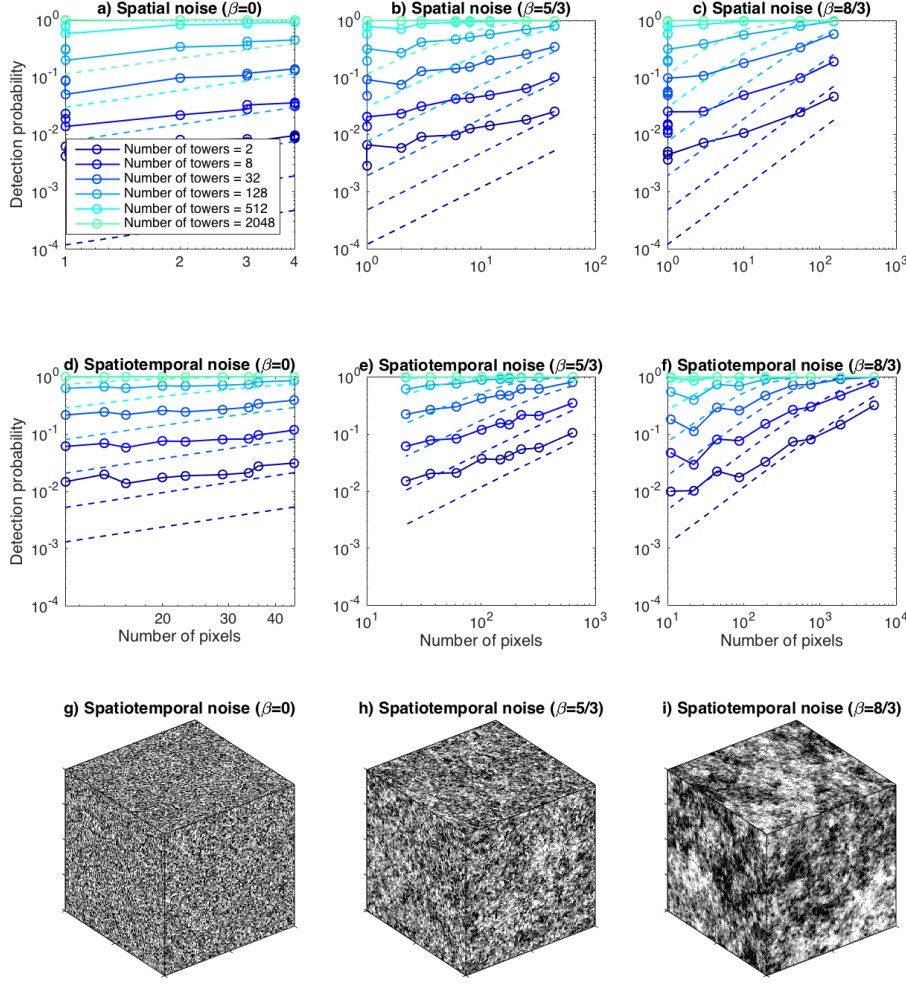

**Figure A5.** Artificial data example considering the actual event detection algorithm. a) Detection probabilities when ignoring the time domain for varying network sizes. In this case, the empirically identified detection rates dramatically overestimate the theoretical detection probabilities. If we induce spatiotemporal correlations in b), and stronger in c) we still find this pattern, but less pronounced for the very large events. This shows that having a large footprint for the event detection algorithm, leads to a dramatic overestimation of the detection rates of small extremes. If the detection rates over space and time are considered, however, the events are not anymore independent due to their temporal autocorrelations and e) and f) reveal lower deviations from the expected detection rates which is a compensating effect of the autocorrelation and event detection method setting. g), f), i) are the data corresponding to results in the columns.





## Appendix B: Analogous example for soil moisture

### B1 On the ISMN

The approach for testing a network design for its capacity to detect extremes is generic by construction. As an additional demonstration we explore the capacity of the **The International Soil Moisture Network ISMN** (http://ismn.geo.tuwien.ac.at/

Dorigo et al., 2011), a steadily growing initiative that comprised collections of soil moisture only. Comparable to FLUXNET there is no specific funding for measurement campaigns, and ISMN crucially depends on the contributions of historical observations by the respective community.

*Methodology*

Direct observations of soil moisture from satellites are available (Liu et al., 2011), but these data still suffer from concate-

nating different data sources. And in fact these transitions make the data set very problematic for detecting extremes – or in other words, extreme event detection may identify the data merging edges. Alternatives are classical drought indicators that can be derived from climatological data only. Here, we rely on the Standardized Precipitation Index (SPI) for detecting extreme events as extracted from SPI and compare it to a random network of the same size (Fig. B1). The SPI is extracted following standard methodology (McKee et al., 1993) from monthly ERA-Interim rainfall data (Dee et al., 2011), using a 3-monthly

aggregation window over the 1979-2011. We us the SPI only for illustration purposes until more robust EO for soil moisture become available, i.e. we assume that low SPI values are proxies for low soil moisture contents.

Further, a local 10th percentile threshold is applied on the SPI time series to flag dry events with subsequent detection of the large connected events. The choice of the local threshold is consistent with the typical meteorological/climatological use of SPI time series. Hence, in contrast to biophysical applications as presented in the main part of the paper, global or regional

thresholds might not be physically meaningful for evaluating the local impacts of climate variables. Since meteorological reanalysis typically operate at much coarser resolution than EO data sets, for the analogous analysis presented here both the spatial and temporal search space are chosen to comprise only the spatially and temporally adjacent voxel (i.e. z = 0.5° and tau= 1 month in the SPI dataset).

To evaluate the ISMN, all station locations and the periods of active data assimilation of each station were used for spatio-

temporal intersection with the SPI extremes in two different setups: Firstly, we consider all stations active only in periods when these stations were collecting data ('dynamic' network); and secondly, a 'static' (counterfactual) situation is taken into account, where all stations are taken as active throughout the entire ERA-Interim period. The comparison was restricted to Europe due to data availability (i.e. most regional networks that form ISMN are operated in Europe (Dorigo et al., 2011)).

*Results*

If we consider the full spatiotemporal intersection we find that only the first five SPI extremes would have affected areas where the ISMN has stations (Fig. B1, red line). Higher ranked extremes are less likely of being detected. An annual random tower placement (gray lines) would have been more efficient in identifying the extremes. In fact the current geographical coordinates $u, v$ would have only reached the potential of a random network if they had been operated without ceasing over the entire monitoring period (blue lines). But that would have implied much more measurement years than the random tower



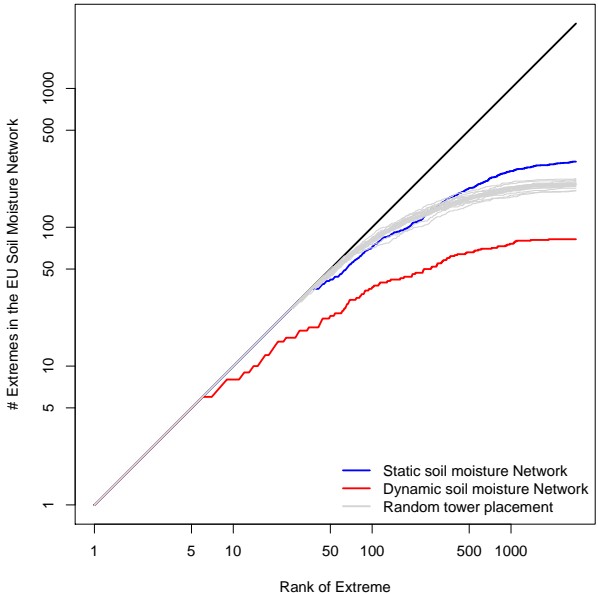

**Figure B1.** International Soil Moisture Network and its capacity to detect SPI extremes in Europe. Again red line showss the reduction of detection capacity due to inactive towers. Randomly placing observatino years in space and time leads to higher detection rates for large extremes, and lower rates for small extremes.

placement. For very high ranks of extremes (the very small events) the continuously operated real-world network would have outperformed the random network. These results are consistent with the results shown in the main paper.

An interesting feature of ISMN is that the network has changed its structure over the last decades to a very large extent. In the eighties, all station locations are confined to East Europe (Fig. B2, upper panel). In the last decade, Western European station

5    networks became active, but both the number and data availability from East European stations was severely reduced (Fig. 11, lower panel). This change in network design materializes strongly in the spatial locations of the detected events: While in the eighties most extremes in East Europe where 'seen' by at least one tower and the detection rates in West Europe were poor, this pattern is reversed in the last decade (Fig. B2). Further, both decades highlight that a static random tower placement is more efficient than the current network, which is explicable by the high degree of site clustering. The importance of maintaining

10   continuous observation alive becomes even more evident if one analyzes the network development over time in more detail (Fig. B3). In conclusion, the complementary analysis presented here substantiates the main paper in that the consideration both the spatial location and the availability of historical data is a crucial element to reconstruct the impacts of extreme events in the recent past.





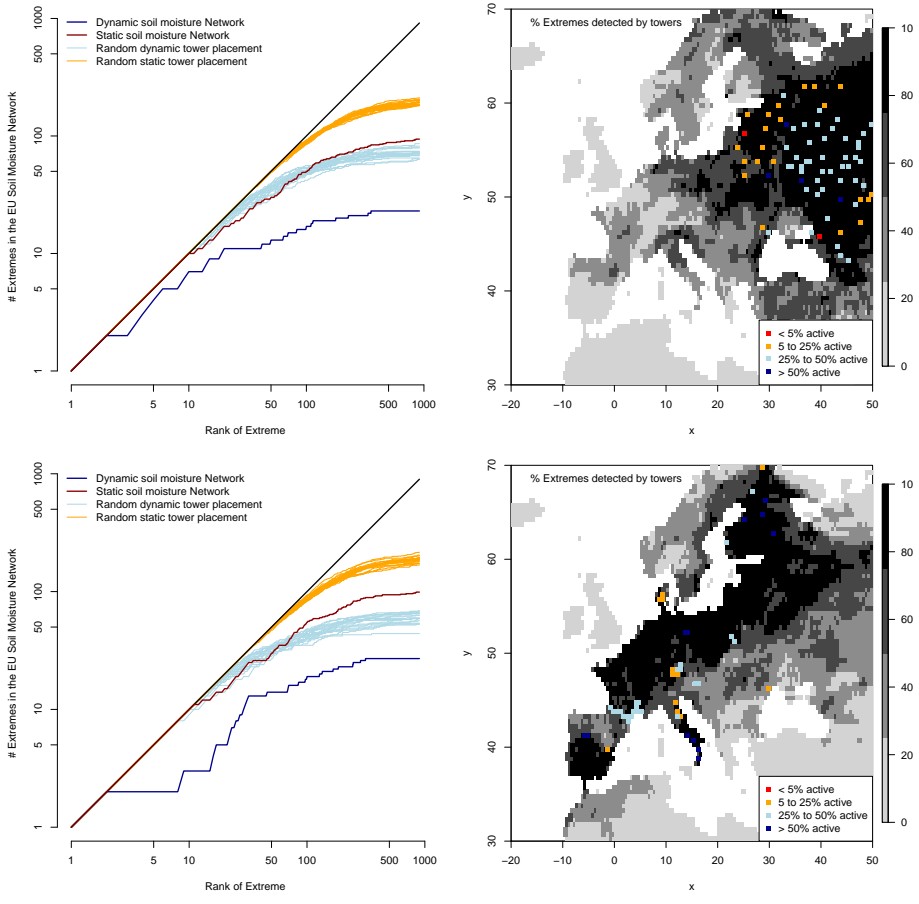

**Figure B2.** International Soil Moisture Network and its capacities to detect SPI extremes in Europe vs. a random network for the 1980ies (upper row) and 2000s (bottom row).





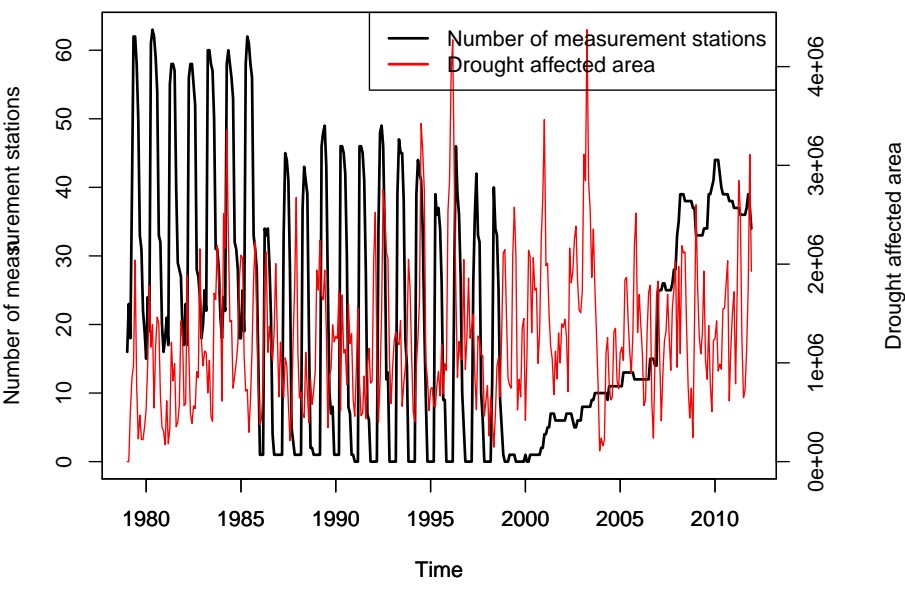

**Figure B3.** Number of stations in the International Soil Moisture Network over time confronted with drought affecte area.



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
