# Peer review of "Detecting impacts of extreme events with ecological in-situ monitoring networks"

_Biogeosciences, 2017_

## Referee Comment (RC1) · G. Bohrer (Referee) · 2 May 2017

The manuscript describes a very well conducted study of the potential to detect extreme events by in situ observation networks. I agree with the analytical approach and the analysis results.

There are however many language and style errors and the manuscript requires a serious editing effort. I provide meticulous editing comments below. These refer to the test until about page 13. There are problems past it, I just didn't have more time.

Minor comments: P3L31 All MODIS datasets have a code (typically starting with three letters, often MOD or MYD with lots of numbers that encapsulate the product type, resolution, return period. . .Pinty references MCD43B3.005, but I think that is what he used to make the next level product). Please list the code of the exact dataset you

used. If it is not from one of the MODIS DAACs and doesn't have a MODIS code, please provide the reference to where you downloaded the data from.

Eq. 1 is trivial and can be removed. The explanation of space-time voxel you provide in P4L3-4 is enough (and can easily be revised so that it doesn't need the equation).

Section 2.2 – please emphasize that you only used the location of each existing flux tower and did not use any of the data collected by the tower. Explaining what types of fluxes and other measurements are done is each site (P4L11-14) is rather confusing and to some degree, misleading

P4L17 Please add the link to Ameriflux, (https://ameriflux.lbl.gov/) and ICOS (https://www.icos-cp.eu/) in the same way you added the links to all other networks you used (fluxnet, euroflux, neon...)

P4L24 Are you sure that the NEON sites "that can be moved for dedicated studies" include EC towers? Did you use any? I think not (for either questions), and if I am correct, you should remove their mention from here as these were not used and create a false impression that they were.

P6L7 Appendix A (or any other appendix) does not include any details of the event detection method. It only contains supplementary figures. Please provide the detailed method description, as it is potentially some of the most exciting and applicable parts of this manuscript.

P10L1 "shown before for the US in Fig. 8." ??? Fig. 8 did not appear yet and nothing was shown for the US

Section 4.1 – please make a conclusive statement (I ended the section confused and wondering)– after all the tests you conducted and results you show in appendix A, is the discrepancy presented in figure 6 explained by the spatial/temporal autocorrelation of the extreme event or the discrepancy must have another explanation. Does it indicate a weakness of random in-situ networks?

[Figure]

Table 1 – lines are strangely discontinuous. Fix it to look like a proper table. In any case, I am not sure that I truly understand the details of what you are trying to convey with this table. Can there be a better way to explain it? Is it necessary and are you using all the categories listed in the table?

P12L3 what is betta? Is it coming from some equation that you did not provide?

Editing comments: P2L27 Today,

P2L29 depths

P2L29 remove "for instance" (already said in L28). Also – collected by (not collected in)

P2L33 And in fact,

P2L34 But it is also shown (cant switch tense in the middle of a sentence)

P3L4 remove repeated "detect"

P3L6 may be of relevant

P3L7 theoretical probabilistic approaches allow us to explain

P3L8 Third, we analyze the detection probabilities. . .

P3L10 I suggest revising to: improved network that could detect and quantify the distribution and effects of extreme events

P3L20 I think you are overdoing it with "plethora" (dictionary meaning - a large or excessive amount of something). I do not think FAPAR supports an excessive amount of studies. Can you simply say "large number of ", or "a large variety of "

P3L28 "but likewise encodes e.g. fire events amongst other extreme reduction of. . ." this sentence is very badly phrased. I was trying to suggest some replacement, but realized that I am not even sure about what you are trying to say here.

P4L6 replace "Firstly" and "Secondly" by "First" and "Then"

P4L9 As an example

P6L8 datasets (one word)

P6L9 i.e. reduce the temporal dimensionality P6L23 replace "firstly" and "Secondly" by "primarily" and "Additionally" . Also – no comma before "that"

P6L31 "the spatial or temporal voxel neighbors" , or "spatially or temporally neighboring voxels"

P8L7 replace "at the center" by "centered on a locally detected extreme event"

P9L15 allows us to answer

P9L15 the hypothetical question whether a certain

P9L17 investigate whether random (similarly, replace "if" by "whether" for indirect questions in many places in the manuscript)

P9L18 remove "amongst other questions" (or rephrase the sentence as an example for other questions that could be studied)

P9L29 across network sizes

P10L3 as black lines

P11L12 within the inline equation, put the n on the left side of the ln to match the format of eq. 4

P11L24 events, which

P13L12,14 replace" firstly" and "secondly" with "(1)" and "(2)"

P14L1 both can be obtained

P14L11 expected detection rates for various

---

## Referee Comment (RC2) · Anonymous Referee #2 · 9 May 2017

This study looks at the detectability of extreme events using existing ecological networks and numerical experiments. I think there is some sound science here but several issues (many are language-related) need to be addressed to make this suitable for publication in BG. Specific comments follow...

P1L10: I'd like some other yardstick to gauge why I might care about the largest 8 vs. largest 39 events. How many sigma departures? Something more than a simple number would help the abstract. Right now the reader has little idea if these are only statistically as opposed to scientifically interesting.

P2L13: To a large extent is unnecessarily vague. Please add some actual information here.

P2L14: Fix the double ecosystem functioning.

P3L3: Please redo your objectives. It's just poorly written, I have no idea where we're going from here–until I re-read this paragraph the third time. That is twice too many. A simple declarative roadmap would be most helpful here.

P4L14: What do you mean considered? Did you use it or not? If I consider traveling to London over the weekend I may or may not go. That is not what you mean to imply here?

P5Fig2: Unclear. Especially the "within each mesh cell" bit. Why are the central dots and assigned percentiles interesting?

P6L1: I'm not sure I see your argument for local vs. global thresholds. Several EO products normalize by location to obviate this issue. And, of course, you do not define "implausible and undesired".

P6L16: How this characteristic FAPAR anomaly is assigned is still a mystery... You never say how the estimation occurs.

P622: If you want clusters why not cluster the MSC directly? And I'm curious how this improves upon other maps of similarity. Why not use MODIS PFTs as is. Or Koeppen-Geiger. What have you gained by this exercise?

P6L11: How well do the top 3 work?

P7Fig3; How is "we may classify an event as extreme in one ecosystem that would be considered part of the normal variability elsewhere" interesting? This is hardly new information! And there are no percentiles here?

P7Fig4: I like this. Good visualization of the sparse network detection problem!

P8L3: Would like an example of z here.

P8L1: Here the reader is confused. You spend much time on the PCA/cluster bit such that your similarity mask allows for spatially non-contiguous blocks. But here we are doing direct neighbors? While I think I understand the thought process here it needs

additional detail. One might wonder why bother with PCA/cluster if you are doing direct neighbors as, for nearby cells, once can reasonably "assume a similar responsiveness to some extreme event " (from P8L11).

P9L2: Did you look at this alternative? As I read on I have the impression that your workflow has so many tunable (hyper)parameters that I am already doubting the results from this one set on offer. Did you do some grid search? Across algorithms and hyperparameters? You can tweak how you arrive at the clusters, how you set thresholds (still unclear), and the z and tau params?

P9L5: Still unclear on the threshold. I like this list of settings. It's a much needed distillation of the methods above. Your grain size concept here is more informative than the above figure.

P10Fig5: Maybe it's a language thing but if b is anomaly while is it labeled as (reduction of m2 with FAPAR = 1? Fix the missing ) in any event. Also, why does 103 get a black line? (I read on and see this point is discussed but do put this [briefly] in the caption as well.)

P10L1: So Fig 8? What about Fig 6 and 7?

P10L8: How is this different. Seems like the same message here. Also, "that one would need to inflate in-situ networks by orders of magnitude to detect small scale events at comparable rates than large-scale extremes" is hardly new insight. We've known this for some time. If the Earth is hit by a large asteroid, a single sensor will be just fine. If several smaller objects rain down more sensors are needed to track them all. You are just restating common sense here.

P10L14: Here you address a departure from an idealized case and invoke spatiotemporal correlation. This is fine but I would outsource this to an Appendix and state in the main text that deviations in detection probabilities relative to theoretical expectations are driven by spatiotemporal structure in the dataframe and move on. This really

breaks the story as written. An alternative is to include this issue in your objectives as another question. It reads tacked on in any event as is.

P13L18: Same above comment applies here. Here you might frame this as a means to more efficiently design a network. This could foreshadow your conclusions a bit better too.

P14Fig7: If caution is warranted, why show it? Also, why are the edges black. The band across Fenno-Scandinavia is odd. Does your workflow simply create such zones at the domain edge?

P15L5: "We therefore ranked the largest 100 events detectable in continental US by their integrated FAPAR anomalies." I still am missing some information on if these are ecologically relevant. Why should a network be configured to detect a class of anomaly (no matter how real, in a numerical sense) that has no impact on land ecosystems? I am not saying that is the case here but you never really address this point. Put another way, an FAPAR departure does not linearly map onto, say, a carbon/water/energy anomaly.

P16Fig8: Percentile ranges from some bootstrap scheme? How many times? I don't recall seeing that. Also, at this point in the paper I see two options. You either explicitly retool this as technical note (that means shortening the paper; this is the easier path forward). *OR* You embrace FLUXNET as is. That is, on Fig8 why not add the other regional networks, the 2015LaThuile? What level of anomaly can we detect with what we have? How does this vary across the globe? Is, say, India better covered than Africa? What can we learn about optimal designed networks and how does that vary based on the type of anomaly we wish to detect. You can do all this, it's the same workflow, FAPAR is global, and tower coordinates are public domain. You could use greenness/NDVI to get back to 1982 and address some methodological shortcomings as well. To be clear, I am not asking you to write the paper I'd maybe like to write. I dislike that in reviews and would not advocate that herein. My point is that as is you

have, in the main, a technical paper. So either embrace the technical note idea (again, the straightforward path) or grow the scope and relevance (this would entail more but would likely be more impactful on the field). As is the paper sits awkwardly between. (In all fairness, some of this thought on my part is motivated by language issues herein.)

FigA4/5: Why does the legend obscure the lines?

AppB: This needs to be better incorporated in the paper. Right now, it really reads as an add-on that yields but marginal insight.

Language: This article suffers from several language faux pas and other overall awkwardness. For example, in P1L11 we read "These finding are". This is of course quite wrong. See the double detect on P3L4. Or, "The paper is based on three main pillars" (same location). This reads more like a Socratic discourse; not so much a BG paper. See FigureA2, specifically "fore" vs. "for". There are many others; some in specific comments. But overall there are too many and I (mostly) stopped after the Introduction. Please have this article proofread by a native (or native-level) speaker of English before resubmission. The language issues are an unneeded distraction from the actual science. One stylistic quibble is that the paper reads like a math exercise in search of a case study. I would have rather read a paper that seeks to improve our detection abilities wrt extremes in land systems.
* * *

---

## Referee Comment (RC3) · N. Brunsell (Referee) · 16 May 2017

Overall comments:

This manuscript addresses how the size and distribution of an observational network impacts the ability to detect the occurrence of extreme events. This is a highly relevant topic, which has not been given sufficient analysis in light of the expected changes in the underlying distributions of these events due to climate change. Thus the manuscript is timely and important.

I do have a few concerns on the methodology, both from questions about the actual analysis and a few suggestions that might help increase the utility of the analysis for a broader readership. I suspect that many of my concerns about the actual analysis can be addressed by clarifications within the text.

For example, I wonder how the impact of the event is actually determined. How does the event intensity, geographic extent and duration relate to the rank? I can imagine more intense events of small geographic extent having a more significant impact than larger extent, longer duration, less intense events. Also, given the carbon and water emphasis of the introduction, the physiological impacts could be significantly different. This should be commented on at some point.

Specifically in Figure 5, does the top line in each subplot represent the same case? So does the middle point in duration, extent and impact lead to the middle of the rank? How would a long, large geographic extent with less 'impact' appear on the rank?

Assuming that the rank and impact aren't a simple linear relationship, this makes me wonder about the scaling analysis with respect to rank. While the analysis itself is interesting, I expect that the detection probability with respect to impact might be more useful to the broader community. If the rank and impact are related, you should state that relationship (same for geographic extent and duration).

It would also be useful to place some of the discussion within the general discussion on how extreme events are classified (i.e. exceedence probabilities, etc.). There is some discussion of the 2003 Europe and the 2012 US drought/heat wave events, it would be interesting to see where these actual events appear in the rank/impact plots of the analysis for perspective given the media/scientific coverage of these events.

More related to the actual analysis, I wonder about the 'clustering' in the PCA space. Why choose an arbitrary mesh and not apply an actual clustering algorithm? What's the benefit of using the PCA approach rather than a normalized/standardized approach on the time series? The benefit stated is with respect to the smaller magnitudes in some bioregions (e.g. semi-arid), but this could be addressed through a proper standardization and the general probability thresholds in the pixel. You could then use a spatial/temporal clustering algorithm that could address the local/regional issue you discuss. To be clear, I'm not suggesting you do that analysis, but some discussion on

why this is necessarily better would be useful for context given the simplicity of that more traditional approach.

What is the role of the underlying resolution (spatial and temporal) of the data on the detection/clustering algorithm. Given the emphasis of the manuscript on the development of the method, these would seem to be important considerations.

In summary, I find the approach interesting and potentially informative, but I think the manuscript needs additional details on the methodology and some more real world applications to illustrate these benefits and help make the authors case on the significance.

There are a number of minor comments/textual issues that should be addressed as well (page/line number):

2/13. Why is satellite capitalized?

2/25. Why is eddy covariance in quotes?

3/6. 'may be A relevant contribution'

3/15. Why is Fraction of Absorbed PAR in quotes?

4/10. Tower sites

4/24. Delete 'also'

What if different extreme events are occurring at the same time in different regions, are they classified as the same event with a larger extent?

9/24: is event size in terms of impact? How is this determined?

10/4: 'largest' in what sense? I could imagine some networks sizes being better at getting largest impact, but perhaps different for largest extent or duration.

Figure 5: how does the FAPAR impact relate to the magnitude of the FAPAR data? Compared to some transformed (zero mean/unit variance) version of the data, it seems

unlikely that 4 orders of magnitude of FAPAR impact (Figure 5b) are all 'extreme'

10/14-16: This would be helpful above prior to the discussion of the figure.

Figures 5 and 6 should have consistent units (e.g. km^2). Figure 6 caption should include description of solid and dashed lines.

13/3: delete one of the 'each'

13/12: Firstly doesn't need to be capitalized

15/1: 'Ameriflux and NEON'

15/16: 'they are not an effective means'

Caption of Figure A2: fore should be for

---

## Author Comment (AC1) · 3 Aug 2017

**Bold text** has are the comments from the reviewers. Standard text is our response. *Italics* are text elements from the revised paper.

**The manuscript describes a very well conducted study of the potential to detect extreme events by in situ observation networks. I agree with the analytical approach and the analysis results. There are however many language and style errors and the manuscript requires a serious editing effort. I provide meticulous editing comments below. These refer to the test until about page 13. There are problems past it, I just didn't have more time.**

Dear Dr. Bohrer:

[Figure]

Thank you very much for your detailed comments on our manuscript. Indeed, none of the coauthors is a native speaker and we apologize for the inconveniences. We have now asked a native speaker to revise the manuscript for language issues and addressed all other comments. Please find below a point-by-point response to your remarks:

**P3L31 All MODIS datasets have a code (typically starting with three letters, often MOD or MYD with lots of numbers that encapsulate the product type, resolution, return period Pinty references MCD43B3.005, but I think that is what he used to make the next level product). Please list the code of the exact dataset you used. If it is not from one of the MODIS DAACs and doesn't have a MODIS code, please provide the reference to where you downloaded the data from.**

We have now revised the paper citing the original MODIS code. The text now reads as follows: *Here we use FAPAR data derived by the JRC-TIP approach (TIP-FAPAR, Pinty et al. 2011), available at 1 km spatial resolution. These estimates are based on the MODIS broadband visible and near-infrared surface albedo products from NASA Collection 5 at 1 km spatial resolution (MCD43B.005, Schaaf et al. 2002, available on demand from co-author T. Kaminski). These satellite data cover the entire surface every 16 days and the data range from 2000 to 2014; in this study we use data covering Europe and the continental US (excluding Alaska).*

**Eq. 1 is trivial and can be removed. The explanation of space-time voxel you provide in P4L3-4 is enough (and can easily be revised so that it doesn't need the equation).**

We agree, this is not worth an equation. We now simply introduce the notation in the text without labeled equation.

**Section 2.2 – please emphasize that you only used the location of each existing flux tower and did not use any of the data collected by the tower. Explaining what types of fluxes and other measurements are done is each site (P4L11-14) is**

**rather confusing and to some degree, misleading**

Yes, we agree and modified the text accordingly.

**P4L17 Please add the link to Ameriflux, (https://ameriflux.lbl.gov/) and ICOS (https://www.icos-cp.eu/) in the same way you added the links to all other networks you used (fluxnet, euroflux, neon...)**

Yes, sorry for the inconsistency - this is now done.

**P4L24 Are you sure that the NEON sites "that can be moved for dedicated studies" include EC towers? Did you use any? I think not (for either questions), and if I am correct, you should remove their mention from here as these were not used and create a false impression that they were.**

Indeed we did not use the translocatable sites and removed any reference to them in the text.

**P6L7 Appendix A (or any other appendix) does not include any details of the event detection method. It only contains supplementary figures. Please provide the detailed method description, as it is potentially some of the most exciting and applicable parts of this manuscript.**

Indeed, reference to "Appendix A" at this point referred to an earlier version of the paper. In response to the reviewers request we not only revised the description of the event detection methods, but also included a more in depth methodological description as Appendix A, which reads as follows:

*In the following we develop a strategy for defining thresholds of regional relevance that are computationally suitable for dealing with high-resolution remote sensing data like the 1 km FAPAR data considered here. Our aim is to find regions of comparable phenology. Our assumption is that the expected seasonal cycle in FAPAR is a good representation of overall phenology, and hence ecosystem type.*

*The first step considers the data set of mean seasonal FAPAR patterns $\mathbf{F} = \{f_{n,s} : \forall n \in 1, \ldots, N; s \in 1, \ldots S\}$, where each point $n$ is pointing to a geographical location $u, v$ and contains the local mean of seasonal observations $s$.*

*In the second step, we use principal component analysis (PCA) to reduce this $S$-dimensional data set. In other words, we seek orthogonal components that represent the main gradient along the covariances of the seasonal cycles. More formally, the covariances of these centered mean seasonal cycles are given as*

$$\mathbf{C} = \mathbf{F}^t \mathbf{F} . \tag{1}$$

*Common patterns of seasonality are identified by first estimating the $k$ leading eigenvectors,*

$$\mathbf{C} E_k = \lambda_k E_k \tag{2}$$

*where $E_k$ the $k$th eigenvector of length $S$, and $\lambda_k$ the corresponding eigenvalue. The scores of the $k$th principal component are given by*

$$A_k = \mathbf{F} E_k . \tag{3}$$

*and $k$ leading $A_k$ can be interpreted as a proxy for the characteristic patterns underlying the mean seasonal cycles across space. Figure REFERENCE TO FIGURE ! visualizes the three leading principal components as an RGB-color composite, revealing a distinct map of European phenological regions.*

*Third, the question is how to identify regions of similar phenology in this continuous space spanned by the principal components. One could use, for instance, some clustering algorithm. However, given the high density of spatial points and the continuous sampling, an equivalent approach is to choose an equidistant grid in the space of the*

*principal components. We choose a very dense grid, such that each cell is as wide as 4% of the range of the first PC. We then define an FAPAR anomaly threshold as a predefined quantile based on the distribution of FAPAR values separately for each grid cell and its 26 neighbours in the space of the leading 3 PCs. This threshold is assigned to all points in the respective grid cell represented herein. This threshold is assigned to the all points represented therein. Figure REF TO FIGURE illustrates this approach in detail.*

*PLACE FIGURE HERE: THIS WAS FIGURE 2 IN THE ORIGINAL SUBMISSION.*

*We have now proposed a FAPAR threshold for each point and can map this threshold back to the geographical space by remapping each point to the known geographical coordinates $u, v$. This is shown in Fig. REF TO FIGURE 3 WHICH WILL BE FIG 2 IN THE RESUBMISSION.*

**P10L1 "shown before for the US in Fig. 8." ??? Fig. 8 did not appear yet and nothing was shown for the US**

This reference to Fig. 8 was inherited from a very old structure of the paper where Fig 8 came first. We have removed this reference here now as it is indeed not helpful.

**Section 4.1 – please make a conclusive statement (I ended the section confused and wondering)– after all the tests you conducted and results you show in appendix A, is the discrepancy presented in figure 6 explained by the spatial/temporal autocorrelation of the extreme event or the discrepancy must have another explanation. Does it indicate a weakness of random in-situ networks?**

Thanks for highlighting these previous inconsistencies in the manuscript. In a revised manuscript, a more consistent and stringent explanation is provided: We restructured Section 4.1: The previous subsection 4.1.2 on Spatiotemporal Correlations is now in the Appendix (where also the artificial simulations are shows); and the explanation why for the real-data case does underestimate detection probabilities for large extremes, but

overestimate detection rates for small extremes is given in Subsection 4.1.1. We have now also tested our explanation in much more detail (see revised manuscript and Appendix B). In fact, the reason for the underestimation of the detection probabilities for large extremes lies in the fact that spatio-temporal correlations lead to a clustering effect: large extremes are more likely found with increasing distance from the boundaries of the domain (or, in other words, the coasts). This effect leads to a higher occurrence probability of a large extreme in the middle of the domain; and due to the fact that the random networks are indeed placed randomly, these networks tend to have a lower chance to detect large extremes because the sampling probability is not adjusted. If these edge effects are small or negligible, the theoretical predictions work very well (as shown in Appendix B). The overestimation of detection rates of small extremes is due to the search radius and explained in Subsection 4.1.

**Table 1 – lines are strangely discontinuous. Fix it to look like a proper table. In any case, I am not sure that I truly understand the details of what you are trying to convey with this table. Can there be a better way to explain it? Is it necessary and are you using all the categories listed in the table?**

The issue of discontinuous lines comes from the LaTeX typesetting. But you are right, finally we don't need all these categories in this very paper. Hence, we have decided to remove it and extend the text where it was introduced.

**P12L3 what is betta? Is it coming from some equation that you did not provide?**

We have now explained this in the text (moved to the appendix B) in more detail as follows:

*The idea is that the Fourier coefficients of artificial data (spatial white noise) are forced to decay as a power law function across frequencies i.e. proportionally to $f^{-\beta}$. An inverse transformation to space yields a correlated data field. If we choose $\beta = 0$, it corresponds to uncorrelated, $\beta = -\frac{3}{5}$ to moderately correlated, and $\beta = -\frac{8}{5}$ to highly correlated data. Hence, $\beta$ is the decay exponent of the Fourier coefficients.* We also

provide reference with respect to these data generation schemes.

**Editing comments:** . . .

The review provided many very helpful suggestions on the text and we have carefully worked through all of them. We acknowledge the reviewer for his efforts!
* * *

---

## Author Comment (AC2) · 3 Aug 2017

**Bold text** has are the comments from the reviewers. Standard text is our response. *Italics* are text elements from the revised paper.

Dear Anonymous Referee,

Many thanks for your detailed review of our manuscript, your work is highly appreciated. Please find below our detailed responses to your comments:

**This study looks at the detectability of extreme events using existing ecological networks and numerical experiments. I think there is some sound science here but several issues (many are language-related) need to be addressed to make this suitable for publication in BG.**

The language was also criticized by reviewer 1. Indeed, none of the authors is a native speaker. Sorry for the inconvenience! In collaboration with a native speaker we have done our very best to revise the paper (not the responses to the reviewers tough). Please note that also Biogeosciences subjects each paper to a language check after acceptance.

**Specific comments follow** . . .

**P1L10: I'd like some other yardstick to gauge why I might care about the largest 8 vs. largest 39 events. How many sigma departures? Something more than a simple number would help the abstract. Right now the reader has little idea if these are only statistically as opposed to scientifically interesting.**

The departures in term of standard deviations are happening locally (e.g. we check each voxel from it's departure from the regionally expected values). However, these "sigma departures" are independent from the total size i.e. affected area or impacts. Hence, we don't understand what the reviewer mean with "How many sigma departures?". The largest events are simply those who matter and the well known fact that extremes in the terrestrial biosphere happen to be power-law distributed naturally leads to a necessity to focus on the x larges events. The numbers 8 vs. 39 sound arbitrary, but are a direct result of comparing the 90% vs. the 50% chances.

**P2L13: To a large extent is unnecessarily vague. Please add some actual information here.**

We have chosen three examples studies and specified their applications. This reads as follows

*Earth observations (EOs), especially satellite remote sensing data, encode relevant information on anomalous ecosystem functioning (Preifer et al. 2012; McDowell et al. 2015). Examples include the exploration of soil moisture anomalies in tandem with climate patterns to understand anomalous vegetation responses (nicolai-Shaw et*

*al. 2017), snow cover induced albedo anomalies with consequences for local climate /Chen et al. 2017), and the impact of weather extremes on vegetation indices to track anomalies in productivity and explain vector-borne disease outbreaks (Anyamba et al. 2014), among many others. The consistent and contiguous spatiotemporal data coverage, and, more importantly, the fact that observations of the land surface typically integrate a plethora of processes, make EO very attractive for detecting extremes affecting the land surface.*

**P3L3: Please redo your objectives. It's just poorly written, I have no idea where we're going from here–until I re-read this paragraph the third time. That is twice too many. A simple declarative roadmap would be most helpful here.**

We agree that this part should be rewritten. We now suggest the following text for the manuscript:

*In this paper we aim to understand the potential of ecological in-situ networks of varying size for monitoring the impact of extreme events. This paper addresses this issue in three steps: 1) We propose an approach for detecting extremes that are of regional relevance. This step is important to avoid a bias toward considering extremes that take place only in high-variance regions, and may be a relevant contribution beyond our application. 2) We explore a series of random networks of varying sizes to explore the expected detection rates. We aim to understand the observed patterns using probabilistic approaches and formulate a theoretical expectation of detection probabilities of extremes. 3) We then analyze the detection probabilities in two real networks (NEON and Ameriflux) and compare these to random networks of identical size. The paper concludes with an outlook on how our remarks could lead to improvements in network design that could be implemented to improve the detection of extreme events.*

**P4L14: What do you mean considered? Did you use it or not? If I consider traveling to London over the weekend I may or may not go. That is not what you mean to imply here?**

Ok, the paragraph is now clearly stating what we used. See also comment by and responses to reviewer 1. The new text starts as follows: *We use the geographical locations of eddy-covariance flux tower networks but to the actual measurements. Our main target is **FLUXNET**, a global collection of eddy covariance data collected (for in-depth descriptions see Baldocchi 2008, 2014, www.fluxdata.org)* . . .

**P5Fig2: Unclear. Especially the "within each mesh cell" bit. Why are the central dots and assigned percentiles interesting?**

What is interesting here is the derivation of the regional threshold. In order to also address the comments by reviewer 1 we now moved this figure to the new appendix A. This allows us to explain the approach in more detail there.

**P6L1: I'm not sure I see your argument for local vs. global thresholds. Several EO products normalize by location to obviate this issue. And, of course, you do not define "implausible and undesired".**

Our point here is to develop a method that is generally applicable. Of course, if a specific EO product is normalized this can be omitted. We argue, however, that many variables don't come with an intrinsic normalization. For instance, the "Normalized Difference Vegetation Index" is constrained between $-1$ and $1$, but may always remain very low in some arid regions, or very high on others. We want to avoid that all values is either of the areas would be classified always as extreme. Hence, we seek a compromise and this compromise needs to be regional. We have now rewritten this section to motivate this approach better (and, of course, avoid writing about "implausible and undesired" effects):

*The question of how to define extreme events in spatiotemporal data cubes is key to the evaluation of the suitability of ecological in-situ networks. One approach could define some global threshold and identifying values exceeding this threshold as potential extremes ("peak over threshold"). Choosing a global threshold setting is suitable when the question is how extremes add up to global anomalies (Zscheischler et al. 2014), i.e.*

*when one is working with the extensive data properties where the target is the integral over space and time. However, the consequence of setting a global threshold is that values that are flagged as potential extremes will occur exclusively in high variance regions. It could happen that one rather flags entiere regions that are extreme compared to others as extremes, while other regions would apparently never face such an event. An alternative would be using only local thresholds (defined over time at each spatial point $x_{uv}$). However, the latter would necessarily lead to an equal spatial distribution of extreme event occurrences. This is also not what we are searching for. We want to define extremes relative to regions that are characterized by a similar ecophysiology e.g. we want to compare each grid cell with grid cells that have a comparable phenology and search for extremes across these geographical locations. However, as our approach should be entirely data driven we refrain from using precomputed definitions of eco-regions.*

**P6L16: How this characteristic FAPAR anomaly is assigned is still a mystery... You never say how the estimation occurs.**

As stated above, we have now added a new appendix A to explain the procedure in detail.

**P622: If you want clusters why not cluster the MSC directly? And I'm curious how this improves upon other maps of similarity. Why not use MODIS PFTs as is. Or Koeppen-Geiger. What have you gained by this exercise?**

As written in appendix A (see response to reviewer 1), we could indeed cluster the MSCs and considered this at the very beginning but we found two complications that we were able to avoid now: 1) Clustering is leading to a non-uniform partitioning of the space spanned by the MSCs and computationally more expensive. 2) it does not allow us to easily assess which are the neighbouring clusters in order to consider these values for the assignment of the thresholds. Our approach has the advantage that we can compute it on a subset of values (rendering it very efficient), and leading

to a uniform grid that allows us to efficiently include the neighboring "clusters" for the estimation of the threshold of the central cell and is controllable in terms of the variance represented and computationally extremely efficient even on this very large data set. MODIS PFTs or Koeppen-Geiger classes would be possible to use, but are very very coarse and don't reflect the details we see in Fig 1. Hence, these classifications would lead to a very coarse thresholding that is by no means comparable to the continuous threshold as shown in Fig 3 (Fig 2 in the revised version of the paper).

Given that reviewer 3 has likewise emphasized this aspect, we have also expanded the discussion to justify our approach. This paragraph read as as follows:

*More specifically, regarding the details of the chosen methodological approach, one may question why we propose simply binning the leading PCs derived from the MSC of our EO. This approach was mainly developed to effectively deal with the very high resolution of the underlying data, seeking a very efficient subgridding approach. One alternative would have been to e.g. cluster the PCs directly. However, besides the computational costs, conventional clustering methods lead to a non-uniform partitioning of the space spanned by PCs. This non-uniform partitioning makes it slightly more complicated to identify neighbouring clusters, which is necessary to stabilize the quantile-based computation of anomaly thresholds. Having an equal meshgrid over the PCs that we can also compute on a subset of MSCs renders the approach very efficient for very large data sets and is completely data adaptive. It was very important for this exercise to have many small classes, in order to compute a very well regionalized anomaly threshold (shown in Fig. ??), which would not have been achievable using classical climate classifications of ecoregions. A more detailed follow-up study should explore the question of how the choice of the various parameters affects the event detection accuracies. A crucial question in this context will be whether one can tune these parameters such that a baseline of events is well detected.*

**P6L11: How well do the top 3 work?**

We guess that the reviewer is asking how much variance we explain with these components? The leading 3 PCs explain more than 70% of the variance in the data.

**P7Fig3; How is "we may classify an event as extreme in one ecosystem that would be considered part of the normal variability elsewhere" interesting? This is hardly new information! And there are no percentiles here?**

We see that there is an error in the caption of the figure that leads to the confusion: The caption was saying "Map of the regionally varying percentiles used for detecting extreme events." but in fact that percentiles (or, more precisely quantiles) are constant for each region as defined in the space of the leading PCs i.e. q = 0.025. What we see here are the regional FAPAR anomaly thresholds that correspond to the quantile. We now added the quantiles to the description of the figure to clarify this. We hope this solves this misunderstanding.

**P7Fig4: I like this. Good visualization of the sparse network detection problem!**

Thank you.

**P8L3: Would like an example of $z$ here.**

As defined in the paper, $z$ is simply a spatial search extend in units of the data e.g. km, or degree. In section 3.1.2 we define the chosen value of z, which is 5km.

**P8L1: Here the reader is confused. You spend much time on the PCA/cluster bit such that your similarity mask allows for spatially non-contiguous blocks. But here we are doing direct neighbors? While I think I understand the thought process here it needs additional detail. One might wonder why bother with PCA/cluster if you are doing direct neighbors as, for nearby cells, once can reasonably "assume a similar responsiveness to some extreme event " (from P8L11).**

Sorry, if this part was not sufficiently well explained. The PCA space is only used to classify homogeneous regions in the data space. For each of these homogeneous

regions we apply the same threshold for defining what an FAPAR anomaly is. This leads to a flagging of values that we consider anomalies beyond a threshold. The result is a binary data array where values can be 1 or 0, i.e. extreme or not. What we do next is to ask if neighbors (defined by the search space) are likewise flagged as 1 or not. We have added a sentence to distinguish these two aspects.

**P9L2: Did you look at this alternative? As I read on I have the impression that your workflow has so many tunable (hyper)parameters that I am already doubting the results from this one set on offer. Did you do some grid search? Across algorithms and hyperparameters? You can tweak how you arrive at the clusters, how you set thresholds (still unclear), and the z and tau params?**

You are indeed right that there are many tunable parameters and we have now added a critical appraisal of this fact to the discussion. But we have actually no objective criterion to define an "optimal parameter set" except that we assume that a quantile of 0.025 is sufficiently "extreme" to identify the values we seek. Our approach detects very prominent heatwaves like the 2003 event in Europe, providing some confidence to the choice of parameters - but even there the question is what would be the correct definition. Given that we cannot optimize the parameters, all choices are inherently arbitrary. The specific resulting values will depend on the choice of parameters (e.g., a less extreme threshold will lead to larger, more connected extreme events). However, we don't expect the specific choice of parameters to have a significant impact on the conclusions of the paper.

**P9L5: Still unclear on the threshold. I like this list of settings. It's a much needed distillation of the methods above. Your grain size concept here is more informative than the above figure.**

The reviewer is right - the threshold should be defined here. We have changed the list as follows to be more precise: *In summary, in this study we used the following settings:*

- *Mean seasonal cycles computed over a time-span from 2001 to 2014.*

- *The first three PCs binned using a grain size of 4% of the range of the first PC.*

- *For each bin in the PC space and its surrounding 26 cells we estimate the quantile $= 0.025$. The FAPAR-anomaly values corresponding to this quantile are assigned as the threshold for the grid cells corresponding to this central bin.*

- *The search space for detecting extreme events is parameterized with $z = 5$ and $\tau = 1$ corresponding here to a search space of $\pm 5$ km and $\pm 16$ days.*

**P10Fig5: Maybe it's a language thing but if b is anomaly while is it labeled as (reduction of m2 with FAPAR = 1? Fix the missing ) in any event.**

This is just a way to quantify the impact: As FAPAR is unitless, it's anomalies are multiplied by the areas of the associated pixels . . . or one could say how much area has an FAPAR value of one. But this is indeed confusing and we omit this now in the new figure labels.

**Also, why does 103 get a black line? (I read on and see this point is discussed but do put this [briefly] in the caption as well.)**

An earlier critique at a conference was that people wanted a "good" number of sites on display and not just the continuum. Hence, we have picked the one which was closest to the 100 to be highlighted. In fact we also refer to this line in the abstract. The fact that it is 103 and 100 is maybe strange, but we also don't see the reason why 100 should be better.

**P10L1: So Fig 8? What about Fig 6 and 7?**

As explained also to reviewer 1, the reference to Fig. 8 was inherited from a very old structure of the paper where Fig. 8 came first. We have removed this reference here now as it is indeed not helpful.

**P10L8: How is this different. Seems like the same message here.**
Fig. A1 (in the revised version B1) is very different in that it shows the detection probability as a function of network size and not as a function of the event size. But of course this has the same message. From the discussions among coauthors we have simply seen that for some people it is more intuitive to think the one way or the other, so we believe it is worth keeping both figures in the paper - one the in the main part, one in the appendix.

**Also, "that one would need to inflate in-situ networks by orders of magnitude to detect small scale events at comparable rates than large-scale extremes" is hardly new insight. We've known this for some time. If the Earth is hit by a large asteroid, a single sensor will be just fine. If several smaller objects rain down more sensors are needed to track them all. You are just restating common sense here.**

We agree that in some way this is common sense - but our question is how to describe this transition? One innovation of the paper is to explain this phenomenon for extremes in the terrestrial biosphere in a very accurate way. We show that estimating the distribution of extremes helps to estimate how fast the decay rate is allowing us to quantify how many towers are need to detect an extreme of a given size x with some probability larger than x%. In view of this argument, would respectfully disagree in removing this sentence from our manuscript.

**P10L14: Here you address a departure from an idealized case and invoke spatiotemporal correlation. This is fine but I would outsource this to an Appendix and state in the main text that deviations in detection probabilities relative to theoretical expectations are driven by spatiotemporal structure in the dataframe and move on. This really breaks the story as written. An alternative is to include this issue in your objectives as another question. It reads tacked on in any event as is.**

We guess that the reviewer is referring here to section 4.1.2 (and not to P10L14, where

we actually discuss the general case - not the departure from it). We agree that this makes the paper quite complicated and we move this part to appendix B, and only keep a one-sentence summary in section 4.1.

**P13L18: Same above comment applies here. Here you might frame this as a means to more efficiently design a network. This could foreshadow your conclusions a bit better too.**

This part is important for us, as it reveals the link between the power-law distribution and rank. But we now add a comment explaining the implications for the network design aspects:

*In other words: gaining insights about the scaling behaviour of the extremes can be used to formulate clear expectations about event detection probabilities of a given rank and size.*

**P14Fig7: If caution is warranted, why show it? Also, why are the edges black. The band across Fenno-Scandinavia is odd. Does your workflow simply create such zones at the domain edge?**

The method has no edge problem. The band at high latitudes comes from low-quality data that are not used in the study. We have, however removed this figure as we also don't see much added value here.

**P15L5: "We therefore ranked the largest 100 events detectable in continental US by their integrated FAPAR anomalies." I still am missing some information on if these are ecologically relevant. Why should a network be configured to detect a class of anomaly (no matter how real, in a numerical sense) that has no impact on land ecosystems? I am not saying that is the case here but you never really address this point. Put another way, an FAPAR departure does not linearly map onto, say, a carbon/water/energy anomaly.**

We agree that an FAPAR anomaly does not linearly map into a carbon/water/energy

anomaly - but still, land-atmosphere fluxes are tightly coupled to biophysical variables like FAPAR. And there are good reasons to at least scrutinize the in-situ observations in depth, when a strong FAPAR anomaly happens. Hence, we respectfully disagree with the reviewer here and the introduction of the paper clearly explains why anomalies in EOs are a relevant means to detect and describe extremes that affect land ecosystems. We have now added a sentence in section 2.1 explain why FAPAR is actually relevant to monitor extremes in the terrestrial biosphere. Give that FAPAR is directly (even if not linearly) coupled to GPP, we can actually use its anomalies as proxies for suspecting anomalies in GPP and other direct observations.

**P16Fig8: Percentile ranges from some bootstrap scheme? How many times? I don't recall seeing that.**

We have explained more explicitly in the caption of the figure that the range comes from 100 random tower placelements - each of the size of the real network to compare with.

**Also, at this point in the paper I see two options. You either explicitly retool this as technical note (that means shortening the paper; this is the easier path forward). \*OR\* You embrace FLUXNET as is. That is, on Fig8 why not add the other regional networks, the 2015LaThuile? What level of anomaly can we detect with what we have? How does this vary across the globe? Is, say, India better covered than Africa? What can we learn about optimal designed networks and how does that vary based on the type of anomaly we wish to detect. You can do all this, it's the same workflow, FAPAR is global, and tower coordinates are public domain. You could use greenness/NDVI to get back to 1982 and address some methodological shortcomings as well. To be clear, I am not asking you to write the paper I'd maybe like to write. I dislike that in reviews and would not advocate that herein. My point is that as is you have, in the main, a technical paper. So either embrace the technical note idea (again, the straightforward path) or grow the scope and relevance (this would entail more but would likely be more**

**impactful on the field). As is the paper sits awkwardly between. (In all fairness, some of this thought on my part is motivated by language issues herein.)**

The reviewer raises multiple aspects in this comment. The overarching question is if this paper should rather be published as "technical note" and not a full research paper. We would like to clarify that we submitted this paper to the "Ideas and perspectives" section because we think that we can offer both. But if the reviewers and/or editors of Biogeosciences consider that our paper is rather a "Technical Note" we could accept this and have no clear preference. This is simply an editorial decision. In fact, we believe that it is not very useful for the key message to extend the analysis to the entire La Thuile sites. Areas like India and Africa have so few sites that we can hardy detect any events there - except if they would be of continental relevance. And an analysis of this kind would indeed lead to another paper. Another question that comes up here is why we use FAPAR and not some data set dating back to 1982. This choice is because the respective long-term EOs originate from multiple sensors and we are still investigating if this would not lead to artifacts in the event detection. We are currently more comfortable using one consistent spatiotemporal dataset.

**FigA4/5: Why does the legend obscure the lines?**

Because of a weird Matlab property: If we could place the legend outside the subfigure, it would dramatically reduce the subfigure size. However, given that the lines behind the legend are – as visible in all other subfigures – just straight lines, the reader does not miss any information.

**AppB: This needs to be better incorporated in the paper. Right now, it really reads as an add-on that yields but marginal insight.**

We have restructured the discussion. This is now the last part, where the soil moisture case is used to substantiate yet another finding: the value of uninterrupted measurement campaigns substantially increases the event detection rates.

**Language: This article suffers from several language faux pas and other overall awkwardness. For example, in P1L11 we read "These finding are". This is of course quite wrong.**

Indeed there were a series of typos in the paper, but we have revised the language of the entire paper now. So we hope that this issue is solved now.

**There are many others; some in specific comments. But overall there are too many and I (mostly) stopped after the Introduction. Please have this article proof-read by a native (or native-level) speaker of English before resubmission.**

The paper was now checked by a native speaker. Biogeosciences offers yet another proofreading service in case the paper will be accepted.

**The language issues are an unneeded distraction from the actual science. One stylistic quibble is that the paper reads like a math exercise in search of a case study. I would have rather read a paper that seeks to improve our detection abilities wrt extremes in land systems.**

We respectfully disagree here: We came across a few basic statistical effects that should be known to the community. We see that we offer a set of statistical tools for quantitative network design, even if we don't actually solve the problem and offer a final solution. Network-design is still a complex optimization task and here we explain how to consider the design problem when the aim is to specifically detect extreme events.

We have carefully worked on all other editing comments. We acknowledge the reviewer for her/his efforts!

---

## Author Comment (AC3) · 3 Aug 2017

**Bold text** are the comments from the reviewers. Standard text is our response. *Italics* are text elements from the revised paper.

**This manuscript addresses how the size and distribution of an observational network impacts the ability to detect the occurrence of extreme events. This is a highly relevant topic, which has not been given sufficient analysis in light of the expected changes in the underlying distributions of these events due to climate change. Thus the manuscript is timely and important.**

Dear Dr. Brunsell

Many thanks for the overall appreciation.

[Figure]

**I do have a few concerns on the methodology, both from questions about the actual analysis and a few suggestions that might help increase the utility of the analysis for a broader readership. I suspect that many of my concerns about the actual analysis can be addressed by clarifications within the text.**

We tried to include the suggested clarifications wherever possible – see our specific responses below.

**For example, I wonder how the impact of the event is actually determined. How does the event intensity, geographic extent and duration relate to the rank? I can imagine more intense events of small geographic extent having a more significant impact than larger extent, longer duration, less intense events. Also, given the carbon and water emphasis of the introduction, the physiological impacts could be significantly different. This should be commented on at some point.**

Yes, we agree that this point is not understandable from the current version of the paper alone and requires reference to papers published earlier by some of the co-authors of this paper. When we introduce Fig. 5 (which in the revision is now Fig 4), we expand the explanation to clarify this point as follows:

*. . .In contrast, investigating the event durations (Fig. 4d) did not reveal such a clear pattern, which could be explained by the fact that we are dealing with a relatively short time series, in which only a few discrete duration classes can be recognized. The fact that global impacts of extreme events in the terrestrial biosphere behave similarly to those at smaller spatial extents is expected because these properties are known to be strongly correlated as shown e.g. in Reichstein et al. (2013). This study also reported that the duration of extreme events is less strongly correlated with their impact, as we would also suspect from Fig. 4.*

**Specifically in Figure 5, does the top line in each subplot represent the same case?**

Yes, the lines represent cases of a given network size, e.g. the darkest blue is the largest random network placed in Europe.

**So does the middle point in duration, extent and impact lead to the middle of the rank?**

We don't exactly understand what the reviewer means with "middle point", but we suspect that the concern is that some rank $x$ derived from an FAPAR impact would not correspond to rank $x$ derived from the spatial extent of the extreme, or duration. This observation is correct, but – as explained above – the surprising fact is that at least extent and impact are highly correlated, such that the ranks would only differ is a few cases. The case for duration is very different tough.

**How would a long, large geographic extent with less 'impact' appear on the rank?**

The impact is defined here as the sum of voxel areas times the FAPAR anomalies. It could happen indeed, that a stronger FAPAR anomaly over a smaller area leads to the same total impact compared to a less intense anomaly that is spread over a larger area. Again, if this would be serious concern we would not see the strong correlations between impact and extents. We work here with the extents as these are very good approximations for the impacts . . . (see also next response)

**Assuming that the rank and impact aren't a simple linear relationship, this makes me wonder about the scaling analysis with respect to rank. While the analysis itself is interesting, I expect that the detection probability with respect to impact might be more useful to the broader community. If the rank and impact are related, you should state that relationship (same for geographic extent and duration).**

Rank and impact are directly related because rank is defined by sorting the events by impact.

**It would also be useful to place some of the discussion within the general dis-**

**cussion on how extreme events are classified (i.e. exceedence probabilities, etc.). There is some discussion of the 2003 Europe and the 2012 US drought/heat wave events, it would be interesting to see where these actual events appear in the rank/impact plots of the analysis for perspective given the media/scientific coverage of these events.**

We have indeed restructured the discussion to consider this aspect in more detail. We now start with a consideration of our event detection method compared to other approaches and referred to the discussion on the known historical events.

**More related to the actual analysis, I wonder about the 'clustering' in the PCA space. Why choose an arbitrary mesh and not apply an actual clustering algorithm? What's the benefit of using the PCA approach rather than a normalized/standardized approach on the time series? The benefit stated is with respect to the smaller magnitudes in some bioregions (e.g. semi-arid), but this could be addressed through a proper standardization and the general probability thresholds in the pixel. You could then use a spatial/temporal clustering algorithm that could address the local/regional issue you discuss. To be clear, I'm not suggesting you do that analysis, but some discussion on why this is necessarily better would be useful for context given the simplicity of that more traditional approach.**

The mesh is so fine (grid size 4% of the length of PC1) that it actually does not affect the resulting regional anomalies. We have tried this in many ways. But we see that there are some concerns regarding the usage of a linear clustering/binning of the PCA space. So yes, we could indeed cluster the MSCs and considered this at the very beginning but we found two complications that we were able to avoid now: 1) Clustering is leading to a non-uniform partitioning of the space spanned by the MSCs and computationally more expensive. 2) it does not allow us to easily assess which are the neighbouring clusters in order to consider these values for the assignment of the thresholds. Our approach has the advantage that we can compute it on a subset

of values (rendering it very efficient), and leading to a uniform grid that allows us to efficiently include the neigoughing "clusters" for the estimation of the threshold of the central cell and is controllable in terms of the variance represented and computationally extremely efficient even on this very large data set. MODIS PFTs or Koeppen-Geiger classes would be possible to use, but are very very coarse and don't reflect the details we see in Fig 1. Hence, these classifications would lead to a very coarse thresholding that is by no means comparable to the continuous threshold as shown in Fig 3 (Fig 2 in the revised version of the paper). Note that this answer was almost identically given to a similar question by reviewer 2.

**What is the role of the underlying resolution (spatial and temporal) of the data on the detection/clustering algorithm. Given the emphasis of the manuscript on the development of the method, these would seem to be important considerations.**

The coarser the resolution, the easier it becomes. Our personal challenge in this paper (compared to earlier studies like the ones by Zscheischler et al. 2013, 2014 or Reichstein et al. 2014) was indeed to deal with a high resolution EO data set. This made it necessary to deal with e.g. a relatively complex search for identifying contiguous extremes. But all these efforts didn't really change the overall picture which has certainly to do with the scaling relationships in the event size distributions that remain unaffected. We now added the following sentence to section . . . to clarify the role of the resolution.

**In summary, I find the approach interesting and potentially informative, but I think the manuscript needs additional details on the methodology and some more real world applications to illustrate these benefits and help make the authors case on the significance.**

We have substantially revised the methodological descriptions and also reorganized the discussion to clarify the relevance of the paper.

**There are a number of minor comments/textual issues that should be addressed**

**as well (page/line number):**

We have addressed all minor comments. Where questions appeared we provide responses below.

**What if different extreme events are occurring at the same time in different regions, are they classified as the same event with a larger extent?**

No, if an event happens to occur as far as z spatial elements away it is considered a different event.

**9/24: is event size in terms of impact? How is this determined? "Size" is the area affected.**

We clarified this here again as follows:

*To better understand expected extreme event detection rates, we initially explore random networks and their hypothetical capability to detect extreme FAPAR reductions. We focus on Europe and vary the network sizes from $n = 5, \ldots, 10000$ sites on a logarithmic scale, asking how many of the detected extremes can be identified for each size class. More precisely, we investigate the probability that an extreme event of a given size $m$ (measured in terms of affected area) will be detected by $n$ hypothetical towers $P(m, n)$. All following analyses are based on repeating the tower placement 100 times per size class. We mimic real site placement by assuming that a tower is not mobile, i.e. it remains active at a given location over the entire period covered by the FAPAR observations.*

**10/4: 'largest' in what sense? I could imagine some networks sizes being better at getting largest impact, but perhaps different for largest extent or duration.**

"Largest" refers here to the integrated FAPAR anomaly and we refer to the ranks of these events. This is explained now in the paper.

**Figure 5: how does the FAPAR impact relate to the magnitude of the FAPAR data?**

**Compared to some transformed (zero mean/unit variance) version of the data, it seems unlikely that 4 orders of magnitude of FAPAR impact (Figure 5b) are all 'extreme'**

Here, we would like to respectfully disagree with the reviewer comment. The voxels are all flagged as extreme or not according to the same regional threshold criterion. The spatiotemporal extends of the contiguous extremes then emerges from the analysis of contiguous extremes and the impacts are resulting from integrating across the contiguous spatiotemporal extremes. As shown in earlier paper (e.g. by Reichstein et al. (2013), Nature; Zscheischler et al. (2013)) these resulting extremes and impacts of extremes are typically power-law distributed. This implies by definition that the size distribution spans across various orders of magnitude.

**10/14-16: This would be helpful above prior to the discussion of the figure.**

We don't exactly understand this point, as the figure is introduced exactly here.

**Figures 5 and 6 should have consistent units (e.g. km$2$). Figure 6 caption should include description of solid and dashed lines.**

The reviewer is absolutely right! The figures have been created by different co-authors. Now, we have recomputed them with the same units. In this context we also found a unit-conversion issue in Fig. 5 (Fig. 4 in the revised paper) that has been corrected.

We have carefully worked on all other editing comments. We acknowledge the reviewer for his efforts!

---

## Author Response (AR1)

**Associate Editor Decision: Publish subject to minor revisions (Editor review)** (14 Aug 2017) by Christopher A. Williams

Comments to the Author:

The response to reviewer comments and the revised version appear to adequately address all of the concerns raised and the paper should be acceptable for publication with final submission of the revised format.

Please note that a LaTeX-diff-file does not make much sense. The rewording and restructuring really changed too much.*Thank you very much for accepting our responses and revised paper. We have now uploaded all files for the final production as a zip file.*

*Please note that a LaTeX-diff-file does not make much sense. The rewording and restructuring really changed too much.*

**Detecting impacts of extreme events with ecological in-situ monitoring networks**

**Miguel D. Mahecha**[1,2,3], **Fabian Gans**[1], **Sebastian Sippel**[1,4], **Jonathan F. Donges**[5,6], **Thomas Kaminski**[7], **Stefan Metzger**[8,9], **Mirco Migliavacca**[1], **Dario Papale**[10,11], **Anja Rammig**[12], **and Jakob Zscheischler**[4]

[1]Max Planck Institute for Biogeochemistry, 07745 Jena, Germany

[2]German Centre for Integrative Biodiversity Research (iDiv), Deutscher Platz 5e, 04103 Leipzig, Germany

[3]Michael Stifel Center Jena for Data-Driven and Simulation Science, 07743 Jena, Germany

[4]Institute for Atmospheric and Climate Science, ETH Zürich, Switzerland

[5]Earth System Analysis, Potsdam Institute for Climate Impact Research, Telegrafphenberg A62, 14473 Potsdam, Germany

[6]Stockholm Resilience Centre, Stockholm University, Kräftriket 2B, 114 19 Stockholm, Sweden

[7]The Inversion Lab, Tewessteg 4, 20249 Hamburg, Germany

[8]National Ecological Observatory Network, Fundamental Instrument Unit, Boulder, Colorado, USA

[9]University of Colorado, Institute for Arctic and Alpine Research, Boulder, Colorado, USA

[10]Department for Innovation in Biological, Agro-Food and Forest Systems, University of Tuscia, Viterbo, Italy

[11]Euro-Mediterranean Center on Climate Change (CMCC), 01100 Viterbo, Italy

[12]Technische Universität München, Hans-Carl-von-Carlowitz-Platz 2, 85354 Freising, Germany

*Correspondence to:* M. D. Mahecha (mmahecha@bgc-jena.mpg.de)

**Abstract.** Extreme hydrometeorological conditions typically impact ecophysiological processes on land. Satellite based observations of the terrestrial biosphere provide an important reference for detecting and describing the spatiotemporal development of such events. However, in-depth investigations of ecological processes during extreme events require additional in-situ observations. The question is whether the density of existing ecological in-situ networks is sufficient for analyzing the impact of extreme events, or what are expected event detection rates of ecological in-situ networks of a given size. To assess these issues, we build a baseline of extreme reductions in the Fraction of Absorbed Photosynthetically Active Radiation (FAPAR), identified by a new event detection method tailored to identify extremes of regional relevance. We then investigate the event detection success rates of hypothetical networks of varying sizes. Our results show that large extremes can be reliably detected with relatively small networks, but also reveal a linear decay of detection probabilities towards smaller extreme events in log-log space. For instance, networks with $\approx$100 randomly placed sites in Europe yield a $\geq 90\%$ chance of detecting the 8 largest (typically very large) extreme events; but only a $\geq 50\%$ chance of capturing the 39 largest events. These findings are consistent with probability-theoretic considerations, but the slopes of the decay rates deviate due to temporal autocorrelation and the exact implementation of the extreme event detection algorithm. Using the examples of AmeriFlux and NEON, we then investigate to what degree ecological in-situ networks can capture extreme events of a given size. Consistent with our theoretical considerations, we find that today's systematically designed networks (i.e. NEON) reliably detect the largest extremes, but that the extreme event detection rates are not higher than would be achieved by randomly designed networks. Spatio-temporal expansions of ecological in-situ monitoring networks should carefully consider the size distribution characteristics of extreme events if the aim is also to monitor the impacts of such events in the terrestrial biosphere.

**1 Introduction**

Many lines of evidence point towards an intensification of certain hydrometeorological extreme events, such as hot temperature extremes or droughts in many regions of the world over the next few

decades (**?**). Consequently, much research focuses on understanding how extreme hydrometeorological events affect ecosystems and their functioning (overviews of the state of research and concepts are given e.g. in **????**). For instance, ecosystem responses could be manifested in extreme anomalies of phenology (**?**), biogeochemical fluxes (**?**), or even in altered ecosystem structure due to induced mortality (**?**). Global analyses of the geographical extent and integrated anomalies of extremes in the terrestrial biosphere reveal that only a very few extremes affect large areas, whereas most events are only of very local relevance (**?**). Nevertheless, the integrated effects of extreme events may have global relevance. For instance, **?** showed that extreme anomalies in gross primary production (GPP) to a large extent explain global inter-annual variability in gross carbon uptake.

Earth observations (EOs), especially satellite remote sensing data, encode relevant information on anomalous ecosystem functioning (**??**). Examples include the exploration of soil moisture anomalies in tandem with climate patterns to understand anomalous vegetation responses (**?**), snow cover induced albedo anomalies with consequences for local climate (**?**), and the impact of weather extremes on vegetation indices to track anomalies in productivity and explain vector-borne disease outbreaks (**?**), among many others. The consistent and contiguous spatiotemporal data coverage, and, more importantly, the fact that observations of the land surface typically integrate a plethora of processes, make EO very attractive for detecting extremes affecting the land surface.

Although EOs enable the detection of extremes in the terrestrial biosphere, a deeper understanding of impacts on ecosystem functioning can be gained from combining EOs with in-situ observations (**??**). In fact, ecological in-situ networks play an increasingly important role in analyzing ecological phenomena and often provide a complementary perspective on natural phenomena to EO (**???**) and complement model analyses (**??**). One prominent example is FLUXNET, with its proven record of advancing our understanding of the functioning of terrestrial ecosystems (**?**). FLUXNET assembles data on the turbulent land-atmosphere exchanges of $CO_2$, $H_2O$, and energy via the eddy-covariance (EC) technique (**??**) as they are collected in regional networks at the country or continent scale (e.g. the pan-European Network Integrated Carbon Observation System ICOS, AmeriFlux, AsiaFlux etc.). Today, many additional networks are operational or are concatenating data from past campaigns. For instance, the International Soil Moisture Network (ISMN) includes a wide range of soil-moisture observations at different depths (**??**); phenological observations are collected in EUROPhen (**?**) or Phenocam (**?**), and one could easily extend this list.

The site distribution in space of ecological in-situ monitoring networks is typically sparse. One obvious and common critique is that networks emerging either as voluntary associations of sites or being constructed on the basis of existing sites (naturally) cannot provide an equitable representation of the world's ecosystems (**?**). And in fact, geographic clustering of sites (**?**) as well as incoherence in their temporal continuity is problematic. However, it has also been shown that the problems of network spatiotemporal representation and the limitations of spatiotemporal extrapolations ("upscaling" *sensu* **???**) are relatively minor compared to the advantages of the sheer size of the network (**?**).

In this paper we aim to understand the potential of ecological in-situ networks of varying size for monitoring the impact of extreme events. This paper addresses this issue in three steps: 1) We propose an approach for detecting extremes that are of regional relevance. This step is important to avoid a bias toward considering extremes that take place only in high-variance regions, and may be a relevant contribution beyond our application. 2) We explore a series of random networks of varying sizes to explore the expected detection rates. We aim to understand the observed patterns using probabilistic approaches and formulate a theoretical expectation of detection probabilities of extremes. 3) We then analyze the detection probabilities in two real networks (NEON and Ameriflux) and compare these to random networks of identical size. The paper concludes with an outlook on how our remarks could lead to improvements in network design that could be implemented to improve the detection of extreme events.

**2 Data**

**2.1 Earth observations, EO**

We required a catalogue of extreme events experienced by terrestrial ecosystems in the past several years to analyze the suitability of in-situ networks for detecting them. To create such a catalogue of extreme impacts, we used extreme negative anomalies of the Fraction of Absorbed Photosynthetically Active Radiation, FAPAR. These values are a dimensionless spatiotemporal indicator of how much solar radiation energy (in the PAR domain) is effectively absorbed by vegetation i.e. converted by photosynthesis (**??**).

FAPAR is considered an "Essential Climate Variable (ECV)" (**?**) because it supports a large variety of studies on the states and variability of the biosphere (*e.g.* **??**) and plays an increasingly important role in the investigation of global biogeochemical cycles (in particular carbon and water fluxes). For instance, FAPAR can be conceptually related to GPP (typically estimated from eddy covariance (EC) tower measurements). This relationship is of the general form $GPP = \varepsilon \times FAPAR \times PAR$, where $\varepsilon$ is some "light use efficiency", and PAR is the "photosynthetically active radiation" (e.g. **?**); one may also include other

limiting factors. Consequently, FAPAR is an important basis for empirical estimates of GPP (**???**) and other relevant ecosystem-atmosphere fluxes *e.g.* evapotranspiration (ET; **?**) or is directly used as input to diagnostic biosphere models (**??**). Given the tight link between FAPAR and land-surface fluxes, this variable has been used in various studies as a reference for monitoring extremes affecting terrestrial ecosystems (**??**).

The temporal variability of FAPAR is influenced by vegetation development, but likewise encodes e.g. fire events and other extreme reductions of FAPAR that are assumed to have a pronounced effect on GPP. Here we use FAPAR data derived by the JRC-TIP approach (TIP-FAPAR, **?**). These estimates are based on the MODIS broadband visible and near-infrared surface albedo products from NASA Collection 5 at 1 km spatial resolution (MCD43B.005, **?**, available on demand from co-author T. Kamins These satellite data cover the entire surface every 16 days and the data range from 2000 to 2014; in this study we use data covering Europe and the continental US (excluding Alaska). In the following we denote this data set as a 3D data cube $\mathbf{X} = \{x_{uvt} : \forall\, u \in 1, \ldots, U; v \in 1, \ldots, V; t \in 1, \ldots, T\}$ where $u$ is the index across the $U$ grid longitudes, $v$ the corresponding index on $V$ latitudes, and $t$ is the index on the $T$ time steps. Each element $x_{uvt}$ is called a voxel and is characterized by a well-defined space-time volume.

**2.2 In-situ networks**

First, we create artificial random in-situ networks in order to systematically study the effects of varying network sizes and as a reference for the analysis of existing networks. Then we analyze existing or recently established in-situ networks for their capability to detect the impacts of extreme events.

We use the geographical locations of eddy-covariance flux tower networks but to the actual measurements. Our main target is **FLUXNET**, a global collection of eddy covariance data collected (www.fluxdata.org; for in-depth descriptions see **??**). FLUXNET is a bottom-up initiative of regional networks which decided to bring their data to a central repository. Hence, there is no systematic sampling design, resulting in unbalanced spatial coverage biased towards central Europe and the contiguous US (**?**). In the US, FLUXNET is mainly composed of the regional network **Ameriflux** https://ameriflux.lbl.gov/ and we use the geographical coordinates of their towers. In Europe, an overview of the most widely used EC can be found in the **European Fluxes database** http://www.europe-fluxdata.eu, which will be partly maintained in the future by **ICOS** https://www.icos-cp.eu. Here, we rely on the site distribution described in the LaThuile data set (**?**).

The National Ecological Observatory Network, **NEON** (http://www.neoninc.org/; **?**) is an initiative to monitor ecosystems of the United States and was constructed using a systematic sampling design chosen to equitably represent the dominant ecoregions across the US. Comparable to Ameriflux, NEON sites are equipped with eddy covariance towers, but also a large suite of additional instrumentation (**?**), and human-based observations are recorded frequently (**?**). We also use the site coordinates of NEON to compare these with Ameriflux in the US.

[Figure]

**Figure 1.** The top three principal components of the mean seasonal cycles of FAPAR over Europe visualized as red (R), green (G), blue (B) channels. The first component accounts for 84% of the variance. The cumulative explained variances in the first two component explain 95% of the variance, and the first three components sum up to 97%. Similar RGB colour combinations indicate comparable mean phenological patterns. These similarities are used to define overlapping regions of comparable phenology. Within each phenological region we estimate suitable and spatially varying thresholds as references for flagging potential extreme reductions in FAPAR.

**3 Methods**

**3.1 Regional extreme event flagging**

The question of how to define extreme events in spatiotemporal data cubes (see eq. 2.1) is key to the evaluation of the suitability of ecological in-situ networks. One approach would be to define some global threshold and identify values exceeding this threshold as potential extremes ("peak over threshold"). Choosing a global threshold setting is suitable when the question is about how extremes add up to global anomalies (**?**), i.e. when one is working with extensive data properties where the target is the integral over space and time. However, the consequence of setting a global threshold is that values that are flagged as potential extremes will occur exclusively in high variance regions, whereas low variance regions will apparently never experience extreme events. An alternative would be using only highly local thresholds (defined over time at each spatial point $x_{uv}$). However, the

latter approach would necessarily lead to an equal spatial distribution of extreme event occurrences, which is also not desirable. We want to define extremes relative to regions that are characterized by a similar ecophysiology i.e. we want to compare each grid cell with other grid cells that have a comparable phenology and search for extremes across these geographical locations. However, as our approach should be entirely data driven, we refrain from using precomputed definitions of ecoregions.

In the following we develop a strategy to define thresholds of regional relevance. This is an attempt to find a compromise between fully local and global thresholding. Our idea builds on the concept of optical types (**?**), as they have been concretely elaborated for EOs by **?**. The key idea offered by them is that similar autocorrelation functions allows to classify ecosystems according to their temporal dynamics (see also **?**). **?** use the leading principal components of the autocorrelation estimated at each pixel across time-lags. We have developed a similar scheme to identify regions in the EOs that are of similar dynamics, but we use mean seasonal cycles instead of the autocorrelation patterns. The rationale of our choice is that want to also maintain differences in amplitude and phasing. The main steps applied for obtaining a regional threshold are the following (for a full description of the regional event detection method see Appendix A):

1. Estimate mean seasonal cycles of the datasets under scrutiny at each grid cell $u, v$. The mean seasonal cycles are centered around a mean of zero.

2. Reduce the temporal dimensionality of the mean seasonal cycles (MSCs) by a principal component analysis such that each principal component (PC) represents a main feature underlying the seasonal cycles. The orthogonal basis for the PCs can be approximated using a random subset of MSCs, rendering the approach very efficient in dealing with this very large data set. Figure 1 shows the first three PCs as an RGB image map for Europe. Although the nonlinearity of color perception by the human eye limits the quantitative informative value of the map, similar colors still represent regions of similar phenological dynamics in FAPAR, so one can gain an impression of environmental heterogeneity in the investigated area.

3. Identify pixels of comparable phenology by binning the scores of the MSCs on the three leading PCs as illustrated in Fig. A1 into bins of equal size. Note that the bins are very small compared to the length of the PC, guaranteeing a very fine binning.

4. Estimate a characteristic FAPAR anomaly threshold in each bin, considering all grid cell $u, v$ belonging to this bin and grid cell $u, v$ in the adjacent bins. Note that in the case of binning the leading 3 PCs, we have all grid cell $u, v$ in 27 bins to estimate an FAPAR anomaly

threshold as a quantile of the anomalies. Figure 2 illustrates the resulting regional threshold of FAPAR anomalies. In southern European ecosystems, smaller negative anomalies of FAPAR (i.e. higher values in Fig. 2) would be used to flag values as potential extremes. The overall geographical pattern suggests that low-variance regions (i.e. arid ecosystems) typically require smaller deviations from the expected variability to be considered abnormal situations.

[Figure]

**Figure 2.** Map of the regionally varying FAPAR threshold used for detecting extreme events. These thresholds are derived within each subregion as defined by the leading PCs of the mean seasonal cycles. The gradient between central and southern Europe indicates that we may classify an event as extreme in one ecosystem that would be considered part of the normal variability elsewhere, i.e. arid ecosystems have lower thresholds of extremeness in FAPAR compared to humid areas.

The rationale behind this approach is primarily that similar mean seasonal cycles indicate which pixels form a "phenological cluster", requiring the application of similar quantiles. Additionally, the identification of these clusters based on the leading PCs avoids complications of an analogous analysis in geographical space where regions of similar phenology might be spatially separated by some barrier like a different land cover type, orography, or a body of water.

**3.1.1 Contiguous spatiotemporal extremes**

Based on the regional extreme threshold (Fig. 2) one may flag individual events as potential ("candidate") extremes. However, these initially flagged values may likewise reflect observational noise. **?** therefore proposed only considering events as extremes if larger geographical areas are synchronously affected or if the extreme persists over some temporal threshold

(a very similar idea was proposed in the context of monitoring droughts range). This idea is realized by identifying clusters in the data cube where the spatial or temporal voxel neighbors are likewise flagged as potential ("candidate") extremes. Each of these clusters is subsequently considered a singular event; for a conceptual illustration see Fig. 3.

[Figure]

**Figure 3.** Conceptual visualization of the presented approach. An extreme occurs over a well-defined spatiotemporal domain (which could be asymmetric as shown here e.g. on the latitude/longitude projection). The rank of an extreme can be determined e.g. by the anomaly integrated by the red voxels, or the maximum spatial extent (gray area), or the duration along the time axis, amongst other properties. Black lines indicate the spatial position and active time of three in-situ measurement stations. In this example, only one site would have coincided with the extreme and would be considered as a potential basis for exploring the in-situ effects of the event.

A critical step of this process is defining the search space around each voxel for detecting potential neighbor extremes that should be concatenated. Throughout this paper we consider the direct neighborhood around a central voxel as follows:

- We define a spatial search space $z$. Two voxels $x_{uvt}$ and $x_{u'v't}$ $(u \neq u'; v \neq v')$ are connected if $|u - u'| \leq z$ and $|v - v'| \leq z$ to obtain a spatial connectivity structure for a given $t$.

- We also define a temporal search horizon $\tau$ from the central voxel to compare $x_{uvt}$ and $x_{uvt'}$ $(t \neq t')$ connecting them if $|t - t'| \leq \tau$.

Visually speaking, we search a square in space and a short line structure in time centered on a locally detected extreme event. Note that a wide range of alternative spatiotemporal connectivity structures could be used, for instance emphasizing the temporal dimension by extending the search space along the $t$-axis. Our choices of $z = 5$ (corresponding to 25km) and $\tau = 1$ (16days) are adjusted *ad-hoc* to the specific properties of the TIP-FAPAR data with its relatively high spatial resolution. By setting $z = 5$ we guarantee that e.g. similar vegetation types (from which we would assume a similar responsiveness to some extreme event) could be concatenated to one extreme, even if these vegetation types are spatially fragmented due to a mosaic of land cover types. In time we search only starting from the central voxel, but given that we do this at each $v, u$ combination, relatively complex spatiotemporal structures are allowed. Each event may consist of a set of voxels with characteristic geometric properties such as the event average or maximum duration across all affected grid cells, or the maximum areal extent. Another interesting property is the average duration of an extreme per affected grid cell. Another way of looking at these events is to integrate the variable anomaly over the voxels affected by an event, and one could also define additional metrics.

**3.1.2 Specific setting for this study**

In summary, in this study we used the following settings:

- Mean seasonal cycles computed over a time-span from 2001 to 2014.

- The first three PCs binned using a grain size of 4% of the range of the first PC.

- For each bin in the PC space and its surrounding 26 cells we estimate the quantile $= 0.025$. The FAPAR-anomaly values corresponding to this quantile are assigned as the threshold for the grid cells corresponding to this central bin.

- The search space for detecting extreme events is parameterized with $z = 5$ and $\tau = 1$ corresponding here to a search space of $\pm 5$ km and $\pm 16$ days.

**3.2 Coinciding in-situ observations and 3D extremes**

In-situ observations typically capture subgrid-level processes or footprints. For the sake of simplicity, here we assume that each point measurement is representative of one pixel $x_{uv}$ [1 km²] and we intersect geographical positions $u$ and $v$ of the in-situ data with the occurrences of 3D extremes. This approach allows us to answer the hypothetical question of whether a certain observation site would have detected an extreme in the past. An intersection considering the time domain as well would allow us to understand if an extreme had a chance of being effectively observed. Along these lines, we can also investigate whether random placement of towers would have improved or deteriorated the capability to detect extreme events.

**4 Results**

**4.1 Random networks**

To better understand expected extreme event detection rates, we initially explore random networks and their hypothetical capability to detect extreme FAPAR reductions. We focus on Europe and vary the network sizes from $n = 5, \ldots, 10000$ sites on a logarithmic scale, asking how many of the detected extremes can be identified for each size class. More precisely, we investigate the probability that an extreme event of a given size $m$ (measured in terms of affected area) will be detected by $n$ hypothetical towers $P(m, n)$. All following analyses are based on repeating the tower placement 100 times per size class. We mimic real site placement by assuming that a tower is not mobile, i.e. it remains active at a given location over the entire period covered by the FAPAR observations.

Figure 4 shows the average detection success rates for the random networks. The ranks $r$ shown in Fig. 4a are derived here from the integrated spatiotemporal FAPAR anomalies (i.e. the total impact); the latter are displayed in Fig. 4b. Across network sizes we find that empirical event detection probabilities increase with event impact. These increases typically follow a straight line in the log-log plot (power-law-like behavior) for small extremes and then level off for very large event sizes. To better illustrate this pattern, we selected the network of size $n = 103$ and display it as black lines in Fig. 4. This specific network size has a $P \gtrsim 90\%$ chance of detecting the 8 largest extreme events (according to the ranks of integrated FAPAR anomaly, see Fig. 4a). This success rate declines rapidly for smaller events, e.g. we have only a $\geq 50\%$ chance of capturing the $r = 39th$ largest event. An analogous pattern is found for the detection probabilities assessed in terms of spatial extents (Fig. 4c). In contrast, investigating the event durations (Fig. 4d) did not reveal such a clear pattern, which could be explained by the fact that we are dealing with a relatively short time series, in which only a few discrete duration classes can be recognized. The fact that global impacts of extreme events in the terrestrial biosphere behave similarly to those at smaller spatial extents is expected because these properties are known to be strongly correlated as shown e.g. in **?**. This study also reported that the duration of extreme events is less strongly correlated with their impact, as we would also suspect from Fig. 4.

A different view on this phenomenon is offered by Fig. C1 showing the detection likelihood for extremes of a given rank $r$ across varying network sizes. Extremes of low rank (i.e. large in impact) need very small networks to be detected with rates near to 100%, whereas high rank events (of small impact) need much larger networks to reach similar detection rates. The detection probability scales linearly in log-log space with network size, indicating that one would need to inflate in-situ networks by orders of magnitude in order to detect small scale events at comparable rates to large-scale extremes.

**4.1.1 Statistical considerations**

The results shown in Fig. 4c are an empirical approach to describe the detection probability of extremes characterized by a given spatial extent $m$ (measured e.g. in terms of the number of pixels or area affected during an event) using a network constructed with $n$ randomly placed towers. In other terms, this figure reports the probability $P(m, n)$ that *at least one* tower detects the extreme and a single extreme event of spatial extent $m$ is detected by a single randomly placed tower with probability

$$p = \frac{m}{m_{\max}}, \tag{1}$$

where $m_{\max}$ is the maximum possible extent $m$ (in our case the maximally affected area across all time steps). However, an equivalent question is the probability that one extreme *is not detected* by any of the $n$ towers. According to the binomial distribution, the latter probability is $(1 - p)^n$, and our estimated probabilities should be described by

$$P(m, n) = 1 - (1 - p)^n$$

$$= 1 - \left(1 - \frac{m}{m_{\max}}\right)^n. \tag{2}$$

This formulation helps explain the parallel decline (linear in log-log) in the detection probabilities for small extremes: We can rewrite Eq. 2 as

$$P(m, n) = 1 - \exp\left(n \ln\left(1 - \frac{m}{m_{\max}}\right)\right) \tag{3}$$

A Taylor expansion of Eq. 3 for a small number of towers $n$ and small event sizes $m/m_{\max}$ (here realized by assuming that $\lfloor n \ln(1 - \frac{m}{m_{\max}})\rfloor \ll 1$) yields

$$P(m, n) \approx -\ln\left(1 - \frac{m}{m_{\max}}\right) n. \tag{4}$$

Further adjusting this formula for small extremes with $\lfloor \frac{m}{m_{\max}}\rfloor \ll 1$ gives

$$P(m, n) \approx \frac{m}{m_{\max}} n, \tag{5}$$

which, in a logarithmic form reads

$$\ln P(m, n) \approx \ln m + \ln n - \ln m_{max}. \tag{6}$$

We expect that this equation explains the empirically identified parallel lines of positive slope in Fig. 4 and compare our empirical findings to this theoretical expectation. Fig. 5 compares the expected and observed detection probabilities. The leveling off of event detection probabilities for large events is indeed theoretically expected,

[Figure]

**Figure 4.** Comparison of average detection rates for randomly placed networks of different sizes in Europe for the period from 2000 to 2014. The color code shows the moderately exponentially increasing size of networks under consideration. Lines show the average percentage of detected events by (a) rank, (b) integrated FAPAR anomaly, (c) affected spatial area, and (d) event duration. The black line shows the case of a hypothetical network of 103 towers.

but the log-linear scaling for small events is expected to be steeper sensu Eq. 2.

In other words: the observed detection probabilities for small extremes are higher than expected, whereas detection probabilities of large extremes are lower in random networks compared to theoretical expectations. Our hypothesis is that these discrepancies are related to the spatiotemporal correlation structure of the extreme events, which is not taken into account in the above theoretical analysis.

[Figure]

**Figure 5.** Comparison of the affected area of extremes (continuous lines are a subset from Fig 4c) and the theoretical expectation according to a binomial distribution and uncorrelated data (dashed lines) for varying network sizes (shown as different colours). Our empirical detection probability is lower than the the theoretical expected ones for large extremes and higher for small extremes. However, the overall pattern of the expected detection probabilities is well captured by the theoretical expectation.

In order to investigate the discrepancy revealed in Fig. 5, we performed a series of simulations using artificial data that are characterized by varying spatiotemporal correlation structures, and compared these to the expected detection rates. The results of these experiments are reported in Appendix B and let us conclude that there are very few effectively independent observations because the extremes are highly autocorrelated in time. Hence, these strong correlations lead to the fact that the largest spatio-temporal extremes tend to occur at some distance from the boundary of the domain (i.e. from the coasts). Because the networks are randomly placed, i.e. without regard to the differentiated occurrence probabilities of large vs. small extremes, this leads to the observed underestimation of detection probabilities for large extremes. A simple thought experiment can intuitively explain this effect: Imagine a landscape that consists of a contiguous, relatively large mainland (e.g. Europe) *and* a number of islands or otherwise disconnected regions (e.g. Great Britain, Ireland, Sicily) that are all far enough from the mainland that spatio-temporal extremes can by definition not be connected, i.e. exceeding the search space $z$. In addition, imagine that the few largest extremes that affect the mainland exceed the size of any of the islands. In this case, any tower randomly placed on an island cannot contribute to detecting large extremes, which intuitively illustrates why not taking into account the effects of autocorrelation and edge effects in our analysis results in overly optimistic theoretical predictions of detection rates based on the binomial distribution for real world landscapes. Contrarily, for medium-sized and small events, the chosen spatial search space of $z = 5$ leads to an overestimation of detection probabilities in the real data as compared to the theoretical predictions. Nonetheless, the theoretical predictions provide an exact expectation under simplified settings (i.e. no boundary effects, and an event search only in directly adjacent grid cells ($z = 1$), see Appendix B); and are thus useful for illustrating and understanding the almost

linear scaling of detection rates and the size of extremes in log-log space.

**4.2 Scaling issues**

One doubt in applying a regional event detection approach was whether key aspects of extreme event distributions would be affected. Occurrence probabilities of extreme events in the terrestrial biosphere have often been reported to follow a power-law of the form $p(m) \propto m^{-\alpha}$ in the tails, i.e. for some values $\geq m_{\min}$ (see **??**, for scaling examples in FAPAR and gross primary production respectively). Using a maximum likelihood estimator as suggested by **??** we analyze the scaling characteristics of contiguous areas affected by extreme events. We find that the event properties follow a power law (see Fig. C3). The probabilities of areas affected by extremes in both areas decline with $\alpha = 1.85 \pm 0.007$ (uncertainties given as standard errors from 1000 bootstrap samples).

Without over-interpreting these patterns (i.e. many processes could lead to the emergence of these power-laws) we consider that this property could be exploited to inform network design issues: According to **?**, and others there are a few considerations pointing in this direction: the expectation value $E[m(r)]$ of an extreme event of rank $r$ (in this formulation, the largest event has rank 1 as in Fig. 4a) has the form

$$E[m(r)] = cr^{-\frac{1}{\alpha-1}}. \tag{7}$$

where $\alpha$ is the scaling exponent, and $c$ is some normalization constant - both can be obtained from a fit to the empirically obtained rank function $m(r)$. Applying Eq. 7 would allow us to study the network detection probability as a function of rank (see Figs. 4a and C1) and we can insert the expressions into Eq. 3:

$$P(m,n) = 1 - \left(1 - \frac{m(r)}{m_{max}}\right)^n \tag{8}$$

$$= 1 - \left(1 - \frac{cr^{-\frac{1}{\alpha-1}}}{m_{max}}\right)^n \tag{9}$$

Furthermore, using the approximated log-log form of the network detection probability (Eq. 7) yields

$$\ln P(m,n) \approx -\frac{1}{\alpha-1}\ln r + 1\ln n + \ln c - \ln m_{max}. \tag{10}$$

This equation may explain the parallel lines for ranks $r$ corresponding to small extreme event extents $m(r)$ (see e.g. Fig.C1). More importantly, it relates the scaling exponent to the expected detection probabilities. In other words: gaining insights about the scaling behaviour of the extremes can be used to formulate clear expectations about event detection probabilities of a given rank and size.

**4.3 Comparing AmeriFlux and NEON**

Our results so far show that random networks may differ somewhat from our expected detection rates for various reasons. But the overarching hypothesis is that even relatively small networks may have a good chance of detecting large scale extreme events. We therefore consider the configuration of real eddy covariance networks. We now focus on the US (continental areas only) instead of Europe. We have two networks with very different histories and therefore configuration: Ameriflux and NEON, and we consider them both together. Again, we compare our results to random networks of equal size.

The starting point for our considerations was whether ecological in-situ networks have effectively been able to detect the most relevant extreme events experienced by land ecosystems due to their network construction, or if these were lucky circumstances. We therefore ranked the 100 largest events detectable in the continental US by their integrated FAPAR anomalies. We then counted the number of events that could have been detected by at least one of the Ameriflux or NEON towers, or, by taking both together (if all towers would have been active over the entire monitoring period). Fig. 6 shows the number of detected events for these three network configurations of NEON, AmeriFlux, and both together, as a function of their rank.

Due to its large network size, AmeriFlux detects many more extremes than NEON (128 vs. 39 sites in the contiguous US, excluding Alaska and islands). Concatenating both networks helps increase the detection rates for small events. Our next question was whether these detection rates are comparable to random networks of the same size. For the case of NEON we find that the median detection rate of randomly designed networks is slightly higher compared to the real network - which still remains above the 2.5%ile. At first glance this is an unexpected finding: we would expect that undesired vicinity may occur by chance in a random network, increasing redundancy among towers in space compared to the very systematic sampling design of NEON (**?**). We conclude here that while the design efforts used in establishing NEON may pay off for certain studies, they are not an effective means to maximize the detection of extremes. This observation again reflects the lack of spatial regularity in the occurrence of extremes.

The equivalent experiment conducted on the AmeriFlux network yields much higher detection rates for the random networks compared to the established network (Fig. 6). We attribute this difference to one particular characteristic of AmeriFlux: many of the sites in this network are co-located on purpose (e.g. to explore spatial heterogeneity or to monitor different disturbance regimes in adjacent and hence climatologically similar ecosystems). Fig. 6 shows that AmeriFlux sites have a relatively high degree of spatial clustering. If the target were to analyze continental extreme events and guarantee monitoring the largest events, the

AmeriFlux configuration would be suboptimal. In other words: the spatial autocorrelation in an ecological in-situ network that was not systematically designed can be outperformed by a random (and hence spatially independent) network.

Another aspect to investigate in this context is concatenating NEON and AmeriFlux (both data sets are intended to be freely available to the research community, Fig. 6 dashed line). Our results show that this approach would marginally increase the detection capacity. One reason for this marginal improvement is again that AmeriFlux and NEON sites are partly geographically co-located and that AmeriFlux—despite of being a bottom-up activity—already has a significant spread across the country that is competitive with a novel network designed for the purpose of capturing large scale extremes.

[Figure]

**Figure 6.** Comparison of the potential of NEON (39 terrestrial sites) and AmeriFlux (128 sites) for detecting extremes defined by varying thresholds in the contiguous continental US (excluding Alaska and islands). The purple dashed line shows a merged AmeriFlux–NEON network. Dashed lines enveloped by a 95 percentile range are detection rates of random networks. The sizes of the random networks correspond to NEON (blue) and AmeriFlux (brown) and summarize 100 repetitions. We also show the 1:1 line, which would correspond to perfect detection performance and is the theoretical limit.

**5   Discussion**

**5.1   Regionalized event detection**

Reliable event detection algorithms are a prerequisite to addressing the question of how effective in-situ networks are for detecting extreme events of a given geographical extent. Our aim here is to classify events as "extreme" if they exceed an anomaly value that is unusual across regions that follow the same main phenological pattern. This contribution could be relevant to other studies beyond the present application. This method has advantages over using a global threshold, which fundamentally changes the obtained picture and leads to a few hotspots of extremes in regions where the data have high variability (for the case of GPP see **?**). The effect of building on regional thresholds to delineate which anomalies should be considered "extreme" (recall Fig. 2) is that we find only very moderate geographical clustering of event occurrences (not shown). From our viewpoint, this is very logical, as there is no reason why relative extremes should preferentially happen in certain regions. Methods of this kind are particularly relevant in times of increasing availability of EOs to detect impacts rather than referring to anomalous observations in the meteorological records, which may or not affect terrestrial ecosystems. In fact, all of the largest extreme events that have had severe impacts on agriculture and human well-being and attracted the attention of the media are well detectable with our approach. Prominent examples are e.g. the 2003 European heat wave (e.g. **?**), the 2010 Russian heat wave (e.g. **?**), or the 2012 US drought (e.g. **?**), which are all easily detectable both from climate records and remote sensing data. However, the smaller the spatial extents become, the more relevant a remote sensing based regional assessment will be. We also expect that a regionalization of this kind could be useful when using more advanced multivariate event detection algorithms (see e.g. **?**) that can tap into the full potential of many EOs.

Regarding the details of the chosen methodological approach, one may question why we propose simply binning the leading PCs derived from the MSC of our EO. This approach was mainly developed to effectively deal with the very high resolution of the underlying data, seeking a very efficient subgridding approach. One alternative would have been to e.g. cluster the PCs directly. However, besides the computational costs, conventional clustering methods lead to a non-uniform partitioning of the space spanned by PCs. This non-uniform partitioning makes it slightly more complicated to identify neighbouring clusters, which is necessary to stabilize the quantile-based computation of anomaly thresholds. Having an equal meshgrid over the PCs that we can also compute on a subset of MSCs renders the approach very efficient for very large data sets and is completely data adaptive. It was very important for this exercise to have many small classes, in order to compute a very well regionalized anomaly threshold (shown in Fig. 2), which would not have been achievable using classical climate classifications of ecoregions. A more detailed follow-up study should explore the question of how the choice of the various parameters affects the event detection accuracies. A crucial question in this context will

be whether one can tune these parameters such that a baseline of events is well detected.

A further argument in favor of our approach was that we rely on a limited number of events detected in a finite time horizon of available satellite data. Monitoring 15 years of extreme events probably does not allow us to conclude anything about the future occurrences of extreme events. In this sense, this study can only be read as a call for (re)considering the density of ecological networks in network design studies. An alternative would be to also consider climate projections and put more emphasis on more "vulnerable" ecoregions. Non-stationary climate and environmental conditions notwithstanding, we have to acknowledge that extremes are too rare to derive a spatial occurrence probability using data from the satellite era only.

**5.2   Relevance for network design**

To the best of our knowledge, there are only a few realized examples of systematically designed in-situ ecological networks. One of the best examples is NEON, which is therefore particularly interesting in the context of this study. The underlying design principle is to cluster environmental conditions and states, including e.g. precipitation, radiation, topography, and water table depth, among others (**?**). These delineated ecoregions are taken to be representative of approximately homogeneous areas in the mean land-climate system state, and yield an equitable representation of land surface processes in upscaling activities (e.g. the spatiotemporal inter- and extrapolation of land-atmosphere fluxes of $CO_2$, $H_2O$, and others **???**) or model-data integration studies (sensu **?**).

Our finding that concatenating NEON and AmeriFlux would have yielded only a minimal increase in detection capacities for extreme events can be understood as a call to avoid co-locating towers in relatively close vicinities - at least when the objective of detecting extreme events is highly relevant. In fact, when the objective is to monitor and understand the impacts of climate extremes on ecosystems, we show here that probability theoretical expectations should be taken into account but would need to be extended to consider temporal autocorrelation as well as the event detection approaches chosen. In our case, the latter had a relatively large footprint ($z = 5$) in order to not miss events that may appear fragmented due to e.g. heterogeneous landscape characteristics. Clearly, one would need to determine such parametric choices depending on the type of extreme events and underlying question.

Nevertheless, we think that the remarks presented here could become useful elements for quantitative network design studies. In our area, earlier considerations in this direction have put their emphasis on reducing the uncertainties for upscaling fluxes from the site level to continental or global flux fields (**?**). Focussing on this first-order question is of course essential, before focussing on detecting rare anomalies. This is also reflected in the alternative methodological avenues that were used for addressing the network design problem. For instance, carbon cycle data assimilation systems (CCDAS; **?**) were very useful for quantitative network design (QND; see, e.g. **??**) i.e. to evaluate real or hypothetical candidate networks in terms of their ability to constrain target quantities of interest. The QND approach within a CCDAS allows the combination of terrestrial, atmospheric and ultimately also oceanic data streams. A key finding so far was that eddy covariance networks with one site per ecosystem type achieve excellent performance. QND studies have also been performed for EO data streams such as column integrated atmospheric $CO_2$ (**??**). But again, none of these studies so far have attempted to unravel the impacts of extreme events on the terrestrial biosphere, which might be a relevant pursuit for subsequent studies.

Overall, this study can be also seen as a prototype. In appendix B we show that analogous studies can be effectively implemented. There we use the International Soil Moisture Network ISMN and detect EO anomalies using a drought indicator. This very brief analysis stresses one additional aspect that we have effectively ignored through the main paper: the importance of keeping network measurements alive over time. Many of the sites have only been active for short monitoring periods, leading to substantial losses in event detection rates. It is the continuously sustained measurement networks that will substantially improve event detection rates in the long-term.

**6   Conclusions**

This study tries to understand to what degree ecological in-situ networks such as AmeriFlux or NEON can capture extreme events of a given size that  affect land ecosystems. We find, for instance, that the  10  largest that have occurred in the US between 2000 and 2014 would  all have been identified with the current networks, offering a good perspective for in-depth site level analyses of these phenomena. Concretely, this finding means that there is a high chance of capturing  major extreme events – beyond the very few  (2-3) prominent events that may receive major media coverage such as the 2003 heatwave in Europe or the 2012 US drought. In general, we find that "large" extreme events could have been detected in a very reliable way,  whereas there was a linear decay of detection probabilities for smaller extreme events in log-log space. We can explain this general behavior with  straightforward considerations in probability theory, but the slopes of the decay rates deviate: While we find lower detection rates for the very large extremes, the opposite is the case for very small extremes. Experiments with artificial networks reveal that these deviations stem both from  autocorrelation issues and the exact implementation of the detection algorithm.

Our original motivation for pursuing this study was the question of whether one could optimize the design of ecological in-situ networks for maximizing the detection rates of extreme events. And indeed, we find some general rules, i.e. when the goal is detecting very large events (i.e. low rank events), network sizes can differ by up to two orders of magnitude but still yield nearly comparable detection rates. Only if the goal was to reliably enhance the detection probabilities of small-scale events would a disproportionate "investment" in large networks be required, which would then also become orders of magnitude more efficient compared to the small networks.

However, any inference on the future spatial occurrence probability of extremes is not tenable based on data from a decade of observation. It is not only data paucity that limits our insights here: quantitative network design is per se non-trivial in a changing world. We find, however, that certain general patterns could be taken into consideration, for instance the fact that event occurrence probabilities are clearly inversely related to detection probabilities on a very well defined and robust scale, and that the power-law distribution of extreme event size seems to have practical relevance for network design purposes.

*Author contributions.* The first three authors equally contributed to analyses presented in this study, J.F.D. helped in deriving the probability-theoretic explanations for the identified patterns, all authors provided substantial input to the design of the study and discussion of the results.

*Acknowledgements.* This study was supported by the European Space Agency with the Support to Science Element STSE "Coupled Atmosphere Biosphere virtual LABoratory project CAB-LAB" see http://earthsystemdatacube.org/ and we thank the EU for supporting the BACI project funded by the Horizon 2020 Research and Innovation Programme under grant agreement 640176; D.P. further thanks the EU for the ENVRIplus project funded in the same programme under grant agreement 654182. The National Ecological Observatory Network is a project sponsored by the National Science Foundation and managed under cooperative agreement by Battelle Ecology, Inc. This material is based upon work supported by the National Science Foundation under the grant DBI-0752017. Any opinions, findings, and conclusions or recommendations expressed in this material are those of the author(s) and do not necessarily reflect the views of the National Science Foundation. J.F.D. thanks the Stordalen Foundation (via the Planetary Boundary Research Network PB.net) and the Earth League's EarthDoc program for financial support. The reviews by Dr. Bohrer, Dr. Brunsell, and Anonymous as well as many suggestions by Dr. Durso have greatly improved the quality of the manuscript.

**Appendix A: Regional event detection**

In the following we develop a strategy for defining thresholds of regional relevance that are computationally suitable for dealing with high-resolution remote sensing data like the 1 km FAPAR data considered here. Our aim is to find regions of comparable phenology. Our assumption is that the expected seasonal cycle in FAPAR is a good representation of overall phenology, and hence ecosystem type.

The first step considers the data set of mean seasonal FAPAR patterns $\mathbf{F} = \{f_{n,s} : \forall n \in 1, \ldots, N; s \in 1, \ldots, S\}$, where each point $n$ is pointing to a geographical location $u, v$ and contains the local mean of seasonal observations $s$.

In the second step, we use principal component analysis (PCA) to reduce this $S$-dimensional data set. In other words, we seek orthogonal components that represent the main gradient along the covariances of the seasonal cycles. More formally, the covariances of these centered mean seasonal cycles are given as

$$\mathbf{C} = \mathbf{F}^t \mathbf{F}. \tag{A1}$$

Common patterns of seasonality are identified by first estimating the $k$ leading eigenvectors,

$$\mathbf{C} E_k = \lambda_k E_k \tag{A2}$$

where $E_k$ the $k$th eigenvector of length $S$, and $\lambda_k$ the corresponding eigenvalue. The scores of the $k$th principal component are given by

$$A_k = \mathbf{F} E_k. \tag{A3}$$

and $k$ leading $A_k$ can be interpreted as a proxy for the characteristic patterns underlying the mean seasonal cycles across space. Figure 1 visualizes the three leading principal components as an RGB-color composite, revealing a distinct map of European phenological regions.

Third, the question is how to identify regions of similar phenology in this continuous space spanned by the principal components. One could use, for instance, some clustering algorithm. However, given the high density of spatial points and the continuous sampling, an equivalent approach is to choose an equidistant grid in the space of the principal components. We choose a very dense grid, such that each cell is as wide as 4% of the range of the first PC. We then define an FAPAR anomaly threshold as a predefined quantile based on the distribution of FAPAR values separately for each grid cell and its 26 neighbours in the space of the leading 3 PCs. This threshold is assigned to all points in the respective grid cellrepresented herein. This threshold is assigned to the all points represented therein. Figure A1 illustrates this approach in detail.

[Figure]

**Figure A1.** Illustration of identification of regions with similar threshold: We define a grid in the space of the leading PCs (geographically shown in Fig. 1), where each mesh width corresponds to 4% of the total min-max range of the first PC. We assign percentile thresholds as calculated over a $3 \times 3 \times 3$ set of mesh elements (shown in orange) and assign these percentiles to the central dots (shown in red). For the sake of clarity, we illustrate the approach only in the space of the leading two PCs.

We have now proposed a FAPAR threshold for each point and can map this threshold back to the geographical space by remapping each point to the known geographical coordinates $u, v$. This is shown in Fig. 2.

**Appendix B: Spatiotemporal correlations**

Fig. 5 reveals a strong discrepancy between theoretical and observed detection probability. Here we investigate this discrepancy further. We generated Gaussian data but introduced varying spatiotemporal correlation structures of different degrees. We followed the approach suggested by **??** to simulate data with a power law power spectrum of some prescribed exponential spectral decay. The method combines an approach for generating spatial fields of a desired correlation structure that likewise have a similar temporal correlation. The idea is that the Fourier coefficients of some artificial data (white noise) are forced to decay as a power law function across frequencies i.e. proportionally to $f^{-\beta}$. An inverse transformation to space yields a correlated data field. If we choose $\beta = 0$, it corresponds to uncorrelated, $\beta = -\frac{3}{2}$ to moderately correlated, and $\beta = -\frac{8}{2}$ to highly correlated data. These artificial datasets are visualized in Fig. B1g-i. We used a simplified event search radius ($z = 1$, $\tau = 1$) and investigate two cases:

1. Ignoring the time domain: In this case, the empirically identified detection rates correspond exactly to the theoretical detection probabilities. This finding reveals that the spatial correlation structure does not explain a deviation from the theoretically expected pattern (compare appendix Figs. B1a—c). This is explained by the fact that, although patterns of extreme anomalies might be correlated in space, the tower placement is still random and for sufficiently sparse networks and relatively contiguous landscapes (i.e. only small edges, no islands, etc.) it has no effect.

2. Considering spatial and temporal correlations: In this case we find a tendency towards lower detection probabilities. This effect becomes more pronounced with larger extremes and spatiotempoal autocorrelation (see appendix Fig. B1d-f) due to a stronger tendency for large spatio-temporal extremes to occur away from the domain's boundaries, thus any tower that is randomly placed close to a boundary would have a disproportionately low chance of detecting large extremes.

However, the approximation of the expected probabilities for the small events is still inconsistent with our empirical finding (recall Fig. 5). Hence, we repeat the artificial experiment using the exact algorithmic settings applied to the FAPAR data: we allow for a tolerance radius ($z \gg 1$, $\tau = 1$) to identify each extreme by a given tower. Again we distinguish the two cases:

1. Ignoring the time domain: Using a large search radius for detecting extremes (which is clearly necessary in real and e.g fragmented landscapes) leads to increased event detection rates. This effect can lead to higher detection rates that exceed the simple statistical expectations as derived from the binomial distribution by several orders of magnitude in the case of small extremes (see appendix Figs. B2a—c).

2. Considering the full spatiotemporal case reduces the discrepancy slightly (i.e. for large events that would be detected anyway), but still results in an overestimation (see appendix Fig. B2d-f). For very large events, the lines may even cross in the case of strongly autocorrelated data.

These numerical experiments highlight some of the issues that need to be considered in evaluating real networks or quantitative network-design: the phenomena we aim to monitor are highly autocorrelated in time, which leads to considerable edge effects for large events. Therefore, theoretically expected detection rates estimated from the binomial distribution are overly optimistic for large events - unless the effects of autocorrelation and edge effects as a consequence for large events are analytically taken into account.

[Figure]

**Figure B1.** Artificial data example. a) Detection probabilities when ignoring the time domain for varying network sizes. In this case, the empirically identified detection rates correspond exactly to the theoretical detection probabilities. If we induce moderate spatiotemporal correlations in b), and stronger ones in c) we still find an excellent fit to the theoretical expectation because we still have relatively sparse networks and the towers are independent samples of the underlying distribution. If the detection rates over space and time are considered, however, the events are no longer independent due to their temporal autocorrelation, and thus the largest extremes tend to cluster towards the center of the domain. Parts e) and f) show these lower detection rates, and 
[revised manuscript text omitted]

Biogeosciences, 13, 4291–4313, doi:10.5194/bg-13-4291-2016, http://www.biogeosciences.net/13/4291/2016/, 2016.

Ustin, S. L. and Gamon, J. A.: Remote sensing of plant functional types, New Phytologist, 186, 795–816, doi:10.1111/j.1469-8137.2010.03284.x, 2010.

Venema, V., Ament, F., and Simmer, C.: A Stochastic Iterative Amplitude Adjusted Fourier Transform algorithm with improved accuracy, Nonlinear Processes in Geophysics, 13, 321–328, doi:10.5194/npg-13-321-2006, http://www.nonlin-processes-geophys.net/13/321/2006/, 2006a.

Venema, V., Theis, S., and Simmer, C.: Online generation of temporal and spatial fractal red noise, Geophysical Research Abstracts, 8, 09 460, 2006b.

Verstraete, M. M., Gobron, N., Aussedat, O., Robustelli, M., Pinty, B., Widlowski, J.-L., and Taberner, M.: An automatic procedure to identify key vegetation phenology events using the JRC–FAPAR products, Advances in Space Research, 41, 1773–1783, doi:10.1016/j.asr.2007.05.066, 2008.

Williams, M., Richardson, A. D., Reichstein, M., Stoy, P. C., Peylin, P., Verbeeck, H., Carvalhais, N., Jung, M., Hollinger, D. Y., Kattge, J., Leuning, R., Luo, Y., Tomelleri, E., Trudinger, C. M., and Wang, Y.-P.: Improving land surface models with FLUXNET data, Biogeosciences, 6, 1341–1359, doi:10.5194/bg-6-1341-2009, http://www.biogeosciences.net/6/1341/2009/, 2009.

Wingate, L., Ogée, J., Cremonese, E., Filippa, G., Mizunuma, T., Migliavacca, M., Moisy, C., Wilkinson, M., Moureaux, C., Wohlfahrt, G., Hammerle, A., Hörtnagl, L., Gimeno, C., Porcar-Castell, A., Galvagno, M., Nakaji, T., Morison, J., Kolle, O., Knohl, A., Kutsch, W., Kolari, P., Nikinmaa, E., Ibrom, A., Gielen, B., Eugster, W., Balzarolo, M., Papale, D., Klumpp, K., Köstner, B., Grünwald, T., Joffre, R., Ourcival, J.-M., Hellstrom, M., Lindroth, A., Charles, G., Longdoz, B., Genty, B., Levula, J., Heinesch, B., Sprintsin, M., Yakir, D., Manise, T., Guyon, D., Ahrends, H., Plaza-Aguilar, A., Guan, J. H., and Grace, J.: Interpreting canopy development and physiology using the EUROPhen camera network at flux sites, Biogeosciences Discussions, 12, 7979–8034, doi:10.5194/bgd-12-7979-2015, http://www.biogeosciences-discuss.net/12/7979/2015/, 2015.

Xiao, J., Chen, J., Davis, K. J., and Reichstein, M.: Advances in upscaling of eddy covariance measurements of carbon and water fluxes, Journal of Geophyical Research – Biogeosciences, 117, G00J01, doi:10.1029/2011JG001889, 2012.

Zscheischler, J., Mahecha, M. D., Harmeling, S., and Reichstein, M.: Detection and attribution of large spatiotemporal extreme events in Earth observation data, Ecological Informatics, 15, 66–73, doi:10.1016/j.ecoinf.2013.03.004, 2013.

Zscheischler, J., Mahecha, M. D., von Buttlar, J., Harmeling, S., Jung, M., Rammig, A., Randerson, J. T., Schölkopf, B., Seneviratne, S. I., Tomelleri, E., Zaehle, S., and Reichstein, M.: Few extreme events dominate global interannual variability in gross primary production, Environmental Research Letters, 9, 035 001, doi:10.1088/1748-9326/9/3/035001, 2014a.

Zscheischler, J., Reichstein, M., Harmeling, S., Rammig, A., Tomelleri, E., and Mahecha, M. D.: Extreme events in gross primary production: a characterization across continents, Biogeosciences, 11, 2909–2924, doi:10.5194/bg-11-2909-2014, http://www.biogeosciences.net/11/2909/2014/, 2014b.